# POQD: Performance-Oriented Query Decomposer for Multi-vector retrieval

**Yaoyang Liu** [1]    **Junlin Li** [2]    **Yinjun Wu** [2]    **Zhen Chen** [3]

## Abstract

Although Multi-Vector Retrieval (MVR) has achieved the state of the art on many information retrieval (IR) tasks, its performance highly depends on how to decompose queries into smaller pieces, say phrases or tokens. However, optimizing query decomposition for MVR performance is not end-to-end differentiable. Even worse, jointly solving this problem and training the downstream retrieval-based systems, say RAG systems could be highly inefficient. To overcome these challenges, we propose Performance-Oriented Query Decomposer (POQD), a novel query decomposition framework for MVR. POQD leverages one LLM for query decomposition and searches the optimal prompt with an LLM-based optimizer. We further propose an end-to-end training algorithm to alternatively optimize the prompt for query decomposition and the downstream models. This algorithm can achieve superior MVR performance at a reasonable training cost as our theoretical analysis suggests. POQD can be integrated seamlessly into arbitrary retrieval-based systems such as Retrieval-Augmented Generation (RAG) systems. Extensive empirical studies on representative RAG-based QA tasks show that POQD outperforms existing query decomposition strategies in both retrieval performance and end-to-end QA accuracy. POQD is available at https://github.com/PKU-SDS-lab/POQD-ICML25.

## 1. Introduction

Dense retrieval retrieves documents by evaluating their similarity scores with user queries (Mitra et al., 2018; Gao & Callan, 2021; Zhao et al., 2024c). It underpins many systems, in particular, retrieval-augmented generation (RAG) frameworks (Karpukhin et al., 2020), where retrieval accuracy is paramount. Multi-Vector Retrieval (MVR) enhances retrieval accuracy by leveraging multiple representations for finer-grained matching (Khattab & Zaharia, 2020). MVR methods, e.g., ColBERT (Khattab & Zaharia, 2020), decompose queries and documents into smaller units, say tokens. For each query token, we identify the most similar document piece to it and calculate their similarity, which is referred to as the *MaxSim operation* in (Khattab & Zaharia, 2020). Such scores are then aggregated across all query tokens as the overall query-document similarity. Compared to standard dense retrieval solutions, this strategy more effectively captures the fine-grained similarities between queries and documents, enhancing performance in information retrieval (IR) tasks (Khattab & Zaharia, 2020; Santhanam et al., 2022b) and retrieval-based systems like RAG (Xu et al., 2024).

In this paper, we aim to develop a new MVR strategy to enhance the performance of arbitrary retrieval-based systems, with a particular focus on RAG systems. Note that traditional MVR approaches, in particular, ColBERT (Khattab & Zaharia, 2020), decompose queries at the token level. However, as revealed in Section 2, decomposing queries into slightly more coarse-grained units, such as phrases, can yield better results for tasks like retrieval-augmented generation (RAG). Furthermore, we observe that the performance of these tasks highly depends on how we decompose queries. Considering that the space of all possible decomposed sub-queries is exponentially large, this thus raises one critical question, i.e., *how can we effectively generate subqueries of arbitrary granularity to optimize the performance of downstream retrieval-based systems?*

Query decomposition has been widely studied in question answering (QA), especially in multi-hop QA. It aims to break down complicated questions into simpler components, allowing Large Language Models (LLMs) to reason step by step, thereby enhancing QA accuracy. Various question decomposition strategies exist, such as (Li et al., 2024), which prompts LLMs with manually crafted prompts for query decomposition. However, as shown in Figure 1, applying the resulting sub-queries to MVR could retrieve an incorrect image, ultimately generating a wrong answer in the

[1]School of Infomation, Renmin University [2]School of Computer Science, Peking University [3]Fundamental Industry Training Center, Tsinghua University. Correspondence to: Yinjun Wu <wuyinjun@pku.edu.cn>, Zhen Chen <zhenchen@tsinghua.edu.cn>.

*Proceedings of the 42nd International Conference on Machine Learning*, Vancouver, Canada. PMLR 267, 2025. Copyright 2025 by the author(s).

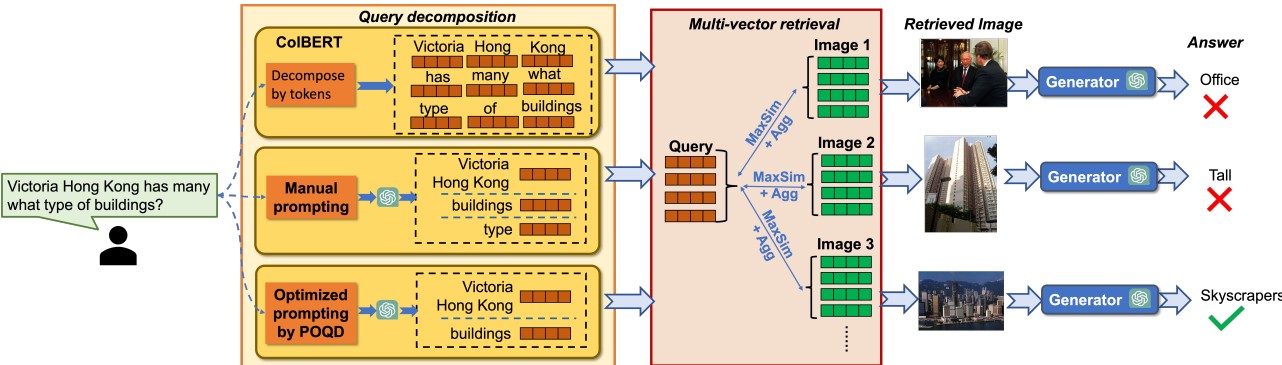

*Figure 1.* Motivating example from ManyModalQA dataset (Hannan et al., 2020). We aim to answer the question "Victoria Hong Kong has many what type of buildings?" using retrieval-augmented generation (RAG). To enhance the retrieval accuracy and thus ensure the answer correctness, we employ Multi-Vector Retrieval (MVR), which decomposes the query into sub-queries and embeds them. MaxSim operations (as defined in (Khattab & Zaharia, 2020)) are then applied to compute similarity scores for retrieval. Traditional query decomposition strategies, which are primarily based on heuristics, such as decomposing by tokens (Khattab & Zaharia, 2020) or by prompting LLMs with manually crafted prompts (Li et al., 2024), often retrieve irrelevant images, thus resulting in incorrect answers. In contrast, we optimize LLM prompts to generate more effective sub-queries, improving QA accuracy. This approach enables MVR to retrieve images of Victoria Harbour with skyscrapers, leading to the correct answer: "Skyscrapers".

RAG-based QA tasks.

To address this issue, it would be ideal to train a model for searching the decomposed sub-queries that can optimize the downstream performance. However, two critical technical challenges arise. First, the search process is non-differentiable, as sub-queries cannot propagate gradients from the downstream performance score. Second, evaluating candidate sub-queries requires training downstream RAG models, which is computationally expensive.

To tackle these two challenges, we proposed Performance-Oriented Query Decomposer (POQD for abbreviation), a novel performance-driven query decomposition framework. To address the non-differentiability issue, we first prompt one LLM to generate decomposed sub-queries for one input query, which can be iteratively refined by an LLM-based optimizer (Yang et al., 2024) to enhance downstream performance. But evaluating a candidate prompt $p$ requires training the downstream model, $\Theta$, with its induced sub-queries. Hence, we propose a training algorithm to alternatively refine the prompt $p$ while only training the model $\Theta$ for a few epochs. Our theoretical analysis confirms the effectiveness of this approach with appropriate hyper-parameter configurations.

Note that such performance optimization process is conducted in a *weakly-supervised* manner since the downstream RAG performance rather than the intermediate retrieval performance is optimized. This strategy is even effective in applications such as Multi-hop QA (Yang et al., 2018), in which the queries are *dynamically generated* during the reasoning process.

We further perform extensive empirical studies to evaluate the effectiveness of POQD on a variety of RAG-based QA tasks, covering both image and text QA tasks. The empirical studies suggest that POQD can outperform the state-of-the-art in both retrieval and QA accuracy by a large margin.

Our contributions can be summarized as follows:

1. We introduce POQD, a novel query decomposition framework that can perform query decomposition for optimizing multi-vector retrieval performance.
2. We design a training algorithm, which alternates between training the downstream RAG models and refining the prompt used for query decomposition. Theoretical analysis demonstrates the effectiveness of this training algorithm with appropriate hyper-parameter configurations.
3. We perform extensive experiments on RAG-based QA tasks, covering both image QA and text QA, which suggests that POQD can outperform the state-of-the-art in both retrieval and QA accuracy by a large margin.

## 2. Motivation

We conduct an in-depth analysis of the motivating example shown in Figure 1 to further motivate our method.

### 2.1. Why does ColBERT fail?

To understand why ColBERT fails in the example shown in Figure 1, we perform MVR with a mini-query "Hong Kong". For this query, ColBERT tokenizes it into two individual tokens, "Hong" and "Kong". For each image, we then perform the MaxSim operation between these two tokens and

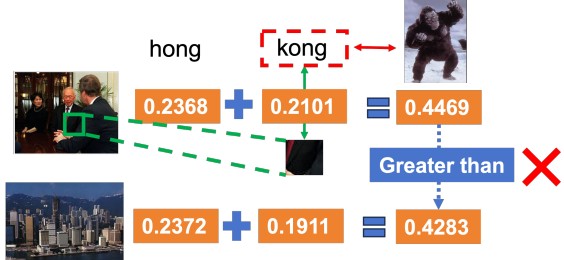

*Figure 2.* Further analysis on the motivating example: the token "kong" is relevant to the photo of a black gorilla-like monster, which is mostly black. Coincidentally, in the photo of Lee Kuan Yew, the patch identified as the most relevant to the token 'kong' is also mostly black.

the fine-grained image patches, i.e., determine the similarity score between one token and its most similar image patch. Such similarity scores are subsequently aggregated across all tokens to obtain the overall query-document similarity score. As depicted in Figure 2, a surprising result emerges: despite its visual irrelevance to Hong Kong, the photo of Lee Kuan Yew, the 1st Prime Minister of Singapore exhibits higher similarity to the mini-query "Hong Kong" than the ground-truth image. A deep investigation suggests that the token "kong" could refer to a black gorilla-like monster, it thus yields unrealistically high similarity to the image patch highlighted with a green bounding box since *both this patch and the figure of "Kong" are mostly black*. This coincidence thus leads to a higher ranking of the Lee Kuan Yew's image than the ground truth. In contrast, by treating "Hong Kong" as a unified phrase and evaluating its similarity to each image, the ground-truth image achieves a higher similarity score than other images. This example thus underscores the necessity of decomposing queries at a slightly coarser-grained level, rather than at the token level for MVR.

### 2.2. Why is it essential to optimize query decomposition?

As mentioned in Figure 1, we can manually craft prompts for LLMs so that they can generate decomposed sub-queries (Li et al., 2024). These sub-queries include critical phrases such as "Victoria Hong Kong" and "building". However, performing MVR with these sub-queries still incorrectly retrieves a less relevant image (see the second row of Figure 1). In comparison to the optimal sub-queries discovered by our solutions, this method generates one extra sub-query "type". This extra sub-query is less informative than the other two sub-queries. Thus incorporating this sub-query into MVR may lead to inaccurate similarity scoring. Therefore, optimizing the query decomposition process by eliminating non-essential sub-queries, such as "type," is crucial for improving retrieval accuracy. Hence, the problem to be addressed is formally defined as follows.

**Problem definition** Given one query $Q = \{c_1, c_2, \ldots, c_m\}$ composed of $m$ tokens, we aim to decompose it into $n$ sub-queries $\{q_i\}_{i=1}^{K}$ in which each $q_i$ is composed of tokens from $\{c_1, c_2, \ldots, c_m\}$. The goal of performing this query decomposition is to maximize the performance of downstream retrieval-based systems.

## 3. Preliminary

### 3.1. Multi-vector retrieval

To evaluate the similarity score between a query $Q$, and a document or image $D$, Multi-vector retrieval first decomposes $Q$ and $D$ into fine-grained pieces, denoted by $\{q_i\}_{i=1}^{K}$ and $\{d_j\}_{j=1}^{m}$, applies MaxSim operation to identify the most similar $d_j$ to each $q_i$ and then aggregates these similarity scores across all $q_i$ with the following formula:

$$\text{SIM}_\theta(Q, D) = \frac{1}{K}\sum_{i=1}^{K} \max_{1 \leq j \leq m} E_\theta(q_i)^\top E_\theta(d_j). \quad (1)$$

As mentioned in Section 2, we primarily study how to optimize the query decomposition process. But how to decompose documents or images may also matter. Therefore, to ensure a fair comparison between different query decomposition strategies, we decompose documents or images in the same way across all baseline methods and POQD. One exception is ColBERT for text retrieval, which is configured to decompose documents into tokens.

### 3.2. Retrieval-Augmented Generation (RAG)

As introduced in Section 1, we primarily study the effectiveness of POQD on RAG tasks. For this task, we aim to optimize the following objective function:

$$\mathcal{L}(\Theta) = -\log(\sum_{D \in D_K} P_\theta(a|Q, D) P_\beta(D|Q)), \quad (2)$$

in which $\Theta = (\theta, \beta)$ and $D_K$ represents the set of Top-$K$ most relevant documents to a query $Q$ according to the similarity score defined in Equation (1). Additionally, $P_\theta$ represents the likelihood of the ground-truth answer $a$, which is parameterized by the generator model parameter $\theta$. Note that this objective function relies on the similarity function defined in Equation (1) to determine the Top-K most relevant documents, thus implicitly dependent on how queries are decomposed. Also, we follow prior studies such as (Barnett et al., 2024), to only train $\theta$ while maintaining the retrieval model, $\beta$, in RAG systems fixed to ensure training efficiency. This is because updating retrieval models usually requires rebuilding indexes and re-embedding corpus, which could be highly time-consuming.

## 4. Methodology

This section starts with the framework overview in Section 4.1, which is followed by illustrating how to generate optimal sub-queries with POQD in Section 4.2 and describing

our end-to-end training algorithm in Section 4.3. We conclude this section with a theoretical analysis of the training algorithm in Section 4.4.

### 4.1. Framework overview

Given a query $Q$, we aim to perform query decomposition by prompting one LLM (referred to as the *Query Decomposer*) with a prompt $p$. Those sub-queries are then employed to perform multi-vector retrieval. Therefore, the quality of the sub-queries produced by the *Query Decomposer* highly depends on the prompt $p$. In light of this, we propose to adopt an LLM-based optimizer (referred to as the *Prompt Optimizer*) to generate $p$ and iteratively refine it for optimizing the downstream performance (see Section 4.2). The pipeline of generating prompt $p$ and producing decomposed sub-queries with this prompt is visualized in Figure 3.

As mentioned in Section 1, the performance to be optimized in our setup would not only depend on $\Theta$, but also depend on the decomposed sub-queries, which is further dependent on the prompt $p$. Hence, the loss function defined in (2) is reformulated as $\mathcal{L}(\Theta; p)$. To optimize this loss function, we proposed an end-to-end training algorithm to jointly optimize $p$ and $\Theta$ (see Section 4.3).

In Section 4.4, we further provide a theoretical analysis of this end-to-end training algorithm. It suggests that with appropriate hyper-parameter configurations, this algorithm can effectively optimize the prompt $p$ and minimize the loss $\mathcal{L}(\Theta; p)$ at a reasonable training cost.

### 4.2. Optimize query decomposition with a fixed $\Theta$

---
**Algorithm 1** Optimize query decomposition

---
1: **Input:** A set of training queries: $\mathcal{Q}^{\text{train}}$, a retrieval-based system parameterized by $\Theta$, the old prompt $p^{\text{old}}$.
2: Initialize the solution-score pairs $LS = [(p^{\text{old}}, \mathcal{L}(\Theta; p^{\text{old}}))]$
3: **while** not converge **do**
4:     Prompt the *Prompt Optimizer* by leveraging $LS$ to generate a new prompt $p$ with **Step 1** of Section 4.2.
5:     Execute **Step 2** to decompose each query in $\mathcal{Q}^{\text{train}}$ by prompting the *Query Decomposer* with $p$.
6:     Evaluate the training loss, $\mathcal{L}(\Theta; p)$, over $\mathcal{Q}^{\text{train}}$ and add $(p, \mathcal{L}(\Theta; p))$ to $LS$
7:     **if** $\mathcal{L}(\Theta; p) - \mathcal{L}(\Theta; p^{\text{old}}) \leq \alpha$ or repeated for $\kappa$ iterations **then**
8:         **Break**
9:     **end if**
10: **end while**
11: **return** $p$ and decomposed sub-queries

---

Given a retrieval-based system with a fixed parameter $\Theta$, we elaborate on how to search optimal decomposed sub-queries in this section. Recall that solving this optimization problem is not differentiable, we overcome this challenge by leveraging Algorithm 1, which iteratively executes the

following two steps. The goals of these two steps are to generate a candidate prompt for the *Query Decomposer* by invoking the *Prompt Optimizer*, and to evaluate the quality of the prompt with the training loss $\mathcal{L}(\Theta; p)$ respectively.

**Step 1:** By following (Yang et al., 2024), the Prompt Optimizer aims to produce a prompt prefix $p_0$, say, "Design a query decomposition framework that..." as shown in Figure 3, which is then concatenated with a fixed prompt template, including the description of the query decomposition task and one input query $Q$, to construct a complete prompt $p$ for the Query Decomposer. Hence, searching optimal $p$ is equivalent to searching optimal prompt prefix $p_0$. The generation of one candidate prompt prefix $p_0$ is conducted by prompting the Prompt Optimizer with two pieces of meta-prompts and a dynamically constructed solution-score pair list (see Figure 3). This list is initially empty and then gradually populated with the pairs of the prompt prefix $p_0$ produced by the Prompt Optimizer and the corresponding training loss $\mathcal{L}(\Theta; p)$ as Algorithm 1 is executed. Intuitively speaking, $p_0$ is regarded as the *solution* to this optimizer while $\mathcal{L}(\Theta; p)$ is viewed as the *score* of this solution.

**Step 2**: To construct the above solution-score pairs, in particular, attaining the training loss $\mathcal{L}(\Theta; p)$ for one candidate prompt $p$, we thus prompt the Query Decomposer with $p$ to generate the decomposed sub-queries for each query in the training set. These sub-queries are then used to perform MVR in the downstream retrieval-based system and evaluate the training loss $\mathcal{L}(\Theta; p)$ on all training queries. Then the pair $(p, \mathcal{L}(\Theta; p))$ is appended to the solution-score pair list as shown in Line 6.

According to (Yang et al., 2024), as more solution-score pairs are included from prior iterations of Algorithm 1, the Prompt Optimizer can gradually refine the prompt $p$ for the Query Decomposer which may produce smaller training loss. In the end, Algorithm 1 terminates if the training loss with the updated prompt, $\mathcal{L}(\Theta; p)$, is at least smaller than that with the initial prompt $p^{\text{old}}$ by $\alpha$ or the while loop is repeated for $\kappa$ iterations (see Line 7 in Algorithm 1).

Note that the Query Decomposer may hallucinate, in particular, the generated sub-queries may contain tokens that do not exist in the input query. To mitigate this, we filter out irrelevant tokens in these sub-queries. The effect of this filtering step is empirically evaluated in Appendix D.3.

### 4.3. End-to-end training algorithm

Note that in Section 4.2, we optimize the prompt with a fixed $\Theta$. Indeed, the sub-queries produced by Algorithm 1 impact the input to $\Theta$, thus motivating the need to further update $\Theta$. As a consequence, we propose an end-to-end training algorithm outlined in Algorithm 2. This algorithm aims to alternatively optimize the prompt $p$ for the query

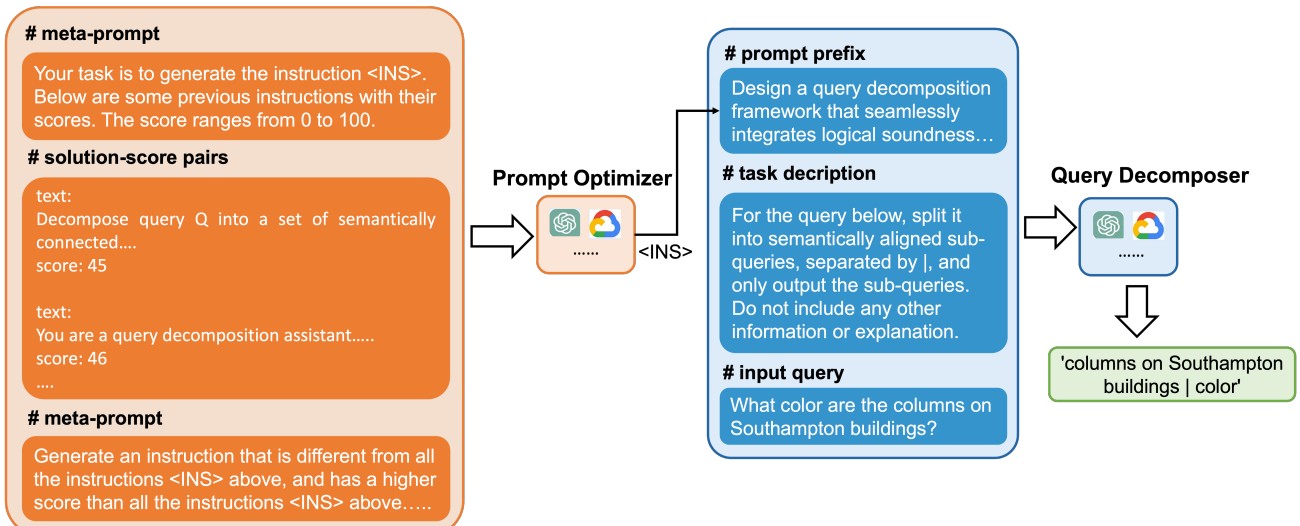

*Figure 3.* The pipeline of generating decomposed sub-queries. This pipeline primarily consists of two LLMs, one serving as the *Prompt Optimizer* while the other one serving as the *Query Decomposer*. The prompt optimizer first takes two pieces of meta-prompts as well as solution-score pairs collected during the execution of Algorithm 1 as input and produces the optimized solution, which is an essential prompt prefix for the *Query Decomposer*. In addition, the query decomposer further incorporates the description of the query decomposition task and one input query in the prompts. The decomposed sub-queries (separated by vertical lines '|') are then generated by the Query Decomposer.

---

**Algorithm 2** Training POQD

1: **Input:** A set of training queries: $\mathcal{Q}^{\text{train}}$, a retrieval-based system parameterized by $\Theta$.
2: Initialize one random $p^{\text{old}}$.
3: **while** not converge **do**
4:     Invoke Algorithm 1 by inputting $p^{\text{old}}$ to obtain a new prompt $p^{\text{new}}$ and optimized sub-queries.
5:     **if** $p^{\text{new}} == p^{\text{old}}$ **then**
6:         **Break**
7:     **end if**
8:     Train $\Theta$ for $\tau$ iterations with optimized sub-queries by minimizing $\mathcal{L}(\Theta; p^{\text{new}})$ with $p^{\text{new}}$ fixed.
9:     $p^{\text{old}} \leftarrow p^{\text{new}}$
10: **end while**
11: Train $\Theta$ until convergence with optimized sub-queries by minimizing $\mathcal{L}(\Theta; p^{\text{new}})$ with $p^{\text{new}}$ fixed.

---

decomposer and train $\Theta$ until convergence.

Note that at each iteration of Algorithm 2, we could optionally train $\Theta$ until convergence given sub-queries produced by Algorithm 1. However, in many retrieval-based systems such as RAG systems, performing full training on $\Theta$ could be highly expensive. For instance, training the RAG model with a large image QA dataset takes up to 1 hour per epoch as revealed in Section 5, which usually needs at least 5 epochs to converge. Hence, in Algorithm 2, we alternatively optimize the prompt and sub-queries in Line 4 and update $\Theta$ for $\tau$ iterations with the optimized sub-queries in Line 8. This is repeated until the prompt cannot be updated any-

more. In the end, we optimize the loss $\mathcal{L}(\Theta; p)$ with a fixed $p$ until convergence, resulting in an optimized parameter $\Theta^*(p)$ (see Line 11 of Algorithm 2). We use the notation $\Theta^*(p)$ to denote its dependency on the prompt $p$.

Note that in Algorithm 1, the training loss gets reduced by $\alpha$ when the prompt is updated from $p^{\text{old}}$ to $p^{\text{new}}$. However, this may not necessarily guarantee decreased training loss at convergence, i.e., $\mathcal{L}(\Theta^*(p^{\text{old}}); p^{\text{old}}) > \mathcal{L}(\Theta^*(p^{\text{new}}); p^{\text{new}})$, which is critical to ensure the optimality of the derived prompt $p^{\text{new}}$. Otherwise, it would be meaningless to update this prompt. Hence, in Section 4.4, we provide a rigorous theoretical analysis to show that the above inequality holds with appropriate $\alpha$ and $\tau$ without hurting training efficiency.

### 4.4. Theoretical analysis

In this sub-section, before formally presenting the theoretical results, we list some essential assumptions below.

**Assumption 4.1** ($\mu$-PL* condition and $L$-smoothness)**.** $\mathcal{L}(\Theta; p)$ satisfies the $\mu$-Polyak-Łojasiewicz star ($\mu$-PL*) condition (Liu et al., 2022) and $L$-smoothness for any $\Theta$ with a given $p$, i.e.,:

$$\|\nabla_\Theta \mathcal{L}(\Theta; p)\|_2^2 \geq \mu \mathcal{L}(\Theta; p), \quad \textbf{(PL* condition)}$$

$$\mathcal{L}(\Theta_2; p) \leq \mathcal{L}(\Theta_1; p) + \nabla_\Theta \mathcal{L}(\Theta_1; p)(\Theta_2 - \Theta_1) + \frac{L}{2}\|\Theta_2 - \Theta_1\|_2^2$$
$$\textbf{(L-smoothness)}$$

Indeed, according to recent theoretical results (Liu et al.),

for pre-trained over-parameterized large language models, if they are fine-tuned with the state-of-the-art optimization method, GaLore (Zhao et al., 2024b), their training loss satisfies the above PL* condition.

**Assumption 4.2** (Bounded loss). For arbitrary $\Theta$ and $p$, the loss $\mathcal{L}(\Theta; p)$ is upper bounded by a constant $M$, i.e.:

$$\mathcal{L}(\Theta; p) \leq M.$$

**Assumption 4.3** (Gradient Descent updates). Suppose updating $\Theta$ in $F(\Theta; p)$ with a fixed $p$ is performed through gradient descent, i.e.,

$$\Theta_{t+1} = \Theta_t - \eta \nabla \mathcal{L}(\Theta_t; p), \tag{3}$$

in which $\eta$ is the learning rate.

Given the above assumptions, the following theorem holds.

**Theorem 4.4.** *Suppose the prompt $p^{old}$ is updated to $p^{new}$ in Line 4 of Algorithm 2, then the following inequality holds:*

$$\mathcal{L}(\Theta^*(p^{old}); p^{old}) - \mathcal{L}(\Theta^*(p^{new}); p^{new}) \geq \alpha - (1 - \frac{\mu}{2L})^{\tau} M,$$

*in which $\mathcal{L}(\Theta^*(p^{old}); p^{old})$ and $\mathcal{L}(\Theta^*(p^{new}); p^{new})$ denote the converged training loss when the prompt is fixed to $p^{old}$ and $p^{new}$ respectively.*

In the above theorem, the term $(1 - \frac{\mu}{2L})$ is a constant between 0 and 1. Therefore, we can configure $\tau$ such that $\alpha - (1 - \frac{\mu}{2L})^{\tau} M$ is a positive value, e.g., $\frac{1}{2}\alpha$ by setting $\tau = \log_{1-\frac{\mu}{2L}}(\frac{\alpha}{2M})$. As mentioned in Section 4.3, we configure $\tau$ as 3 by default which strikes a balance between the training efficiency and performance as empirically verified by Section 5.

This theorem states that with appropriate $\tau$ and $\alpha$, Algorithm 2 can effectively optimize the prompt $p$ for decomposing sub-queries at a reasonable training cost, thus achieving superior downstream performance than that by employing other query decomposition strategies. The complete proof of this theorem is provided in Appendix A.

# 5. Experiments

## 5.1. Experimental setup

**Baseline**   We compare POQD against the following query decomposition methods from prior studies:

- Conventional **dense retrieval** encodes each query and document with one single embedding.
- **ColBERT** (Khattab & Zaharia, 2020) which decomposes queries into individual tokens.
- **Supervised Query Decomposition** (**S-QD** for short): A series of works (Xue et al., 2024; Yang & Zhu, 2021; Zhou et al., 2022; Zhu et al., 2023; Guo et al., 2022) train a sequence-to-sequence model in a supervised manner

to generate decomposed sub-questions for each question. We follow (Zhou et al., 2022; Zhu et al., 2023; Guo et al., 2022; Wu et al., 2024a) to fine-tune Llama3.1-8B with StrategyQA dataset (Geva et al., 2021b) which contains human-annotated sub-queries.

- **Unsupervised Query Decomposition** (**U-QD** for short): This aims to train a query decomposition model in an unsupervised manner. The representative method is OUNS (Perez et al., 2020) which aims to identify sub-queries that are similar to original questions but also diverse enough.
- **In-Context Learning-based Query Decomposition** (**ICL-QD** for short): Some recent works (Li et al., 2024; Pereira et al., 2023; Niu et al., 2023; Ye et al., 2023; Xue et al.; Wu et al., 2024b; Chen et al., 2024; Bhattacharya et al., 2023) prompt LLMs to perform in-context learning for query decompositions with manually crafted prompts. These prompts are included in Appendix D.1.
- **In-Context Learning with Feedback for Query Decomposition** (**ICLF-QD**): Some recent works (Qi et al.; Gao et al., 2024; Sidhoum et al., 2024) improve ICL-QD by providing feedback to LLMs regarding the quality of the decomposed sub-queries. In particular, we follow (Qi et al.) to evaluate whether a sub-query is relevant to the retrieved document or not. This is for determining whether to further decompose this sub-query. The prompts used in this method are included in Appendix D.1.

**Datasets and models**   We employ **Web Questions** (**WebQA**) (Berant et al., 2013; Chang et al., 2021), **MultiModalQA** (Talmor et al.), **ManyModalQA** (Hannan et al., 2020) and **StrategyQA** (Geva et al., 2021a) dataset for experiments. Among these datasets, the former three include questions requiring retrieval from multi-modal data. We focus on two RAG-based QA tasks throughout the experiments, i.e., image QA and text QA. For image QA, we select only questions requiring image retrieval from WebQA, MultiModalQA, and ManyModalQA. For text QA, we select only questions requiring text documents from all of these four datasets. Notably, StrategyQA is used for multi-hop QA, while the others only support single-hop QA.

Regarding the retrieval process, it is critical to determine which embedding model to use. For text QA, we employ the Sentence-Bert model (Reimers, 2019) by default for encoding sub-queries and corpus for other baseline methods as well as POQD. On the other hand, for image QA, the CLIP model (Radford et al., 2021) is employed as the default model for embedding text queries and image corpus. In Section 5.3, we further perform ablation studies on the retrieval model. But note that ColBERT and its counterpart for image retrieval, ColPali (Faysse et al., 2024), have their own encoding models. Hence, we report the results of two versions of ColBERT, one taking its own embedding model (denoted by ColBERT-orig) while the other leverages the

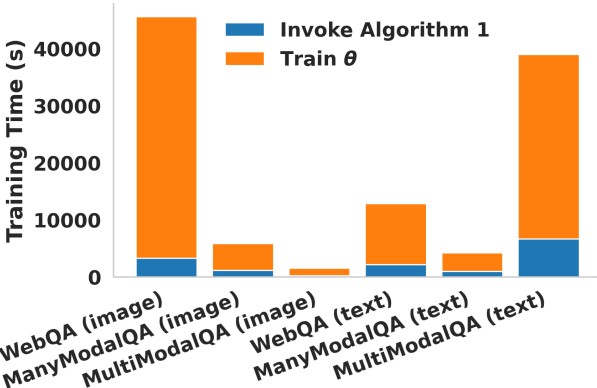

*Figure 4.* Overall training time of Algorithm 2

same default embedding model as others.

For the generator models, we leverage Llama3.1-8B (Dubey et al., 2024) and Llava-v1.5-7B (Liu et al., 2024) as generators for single-hop text QA and image QA, respectively. In the experiments, only these generator models are fine-tuned while keeping the retrieval models frozen. On the other hand, regarding multi-hop text QA, i.e., StrategyQA dataset, we follow the state-of-the-art (Xu et al., 2024) to utilize the frozen GPT-4 (Achiam et al., 2023) model and merely replace its default retrieval method with baseline methods and POQD.

Throughout the experiments, the default values of $\alpha$, $\tau$ and $\kappa$ are configured as 0.02, 3 and 5, respectively. Regarding the configuration for the retrieval process, we retrieve the Top-1 most relevant images and the Top-2 most relevant documents in the image QA and text QA tasks, respectively. More details on experimental setup are provided in Appendix B, which include how to decompose and embed documents or images.

### 5.2. Quantitative results

**Performance analysis** We perform end-to-end RAG training on the QA datasets introduced in Section 5.1. For this experiment, we not only report the end-to-end QA accuracy in Table 2 but also compare the ground-truth relevant documents or images against the retrieved ones by POQD and baseline methods in Table 1. Regarding the retrieval accuracy metric, we report Hit@1 and Hit@2 (see the formal definition in (Croft et al., 2010)) for image QA and text QA, respectively, since the Top-1 relevant image and Top-2 relevant text documents are retrieved in these two tasks, respectively. Note that since StrategyQA is primarily used for multi-hop QA in which the queries for retrieving documents are dynamically generated during the reasoning process, the ground-truth relevant documents are thus not available for this dataset. Hence, we do not report the retrieval accuracy for this dataset.

As Table 1 and Table 2 suggest, POQD outperforms all baseline methods in both the retrieval performance and end-to-end QA accuracy by a large margin across all datasets. Notably, the retrieval accuracy is increased by up to 5.28% (see the last column in Table 1) while POQD boosts QA accuracy by up to 12.61% (see the MultiModalQA column under Image QA in Table 2). This thus indicates that performing multi-vector retrieval with the sub-queries derived by POQD, can enhance the retrieval performance, and consequently the QA accuracy. Note that POQD consistently beats both ColBERT and ColBERT-orig, thus indicating the poor performance of ColBERT regardless of the underlying embedding model.

**Time analysis** We further analyze both the training time and inference time of POQD for the RAG-based QA pipeline. First, regarding the training time, we record the overall running time of Algorithm 2 on all datasets except StrategyQA. StrategyQA is excluded since, as noted earlier, the generator model is not fine-tuned for this multi-hop QA dataset. The results, presented in Figure 4, also decompose the total running time into two components: the time for invoking Algorithm 1 to optimize the prompt $p$ in $\mathcal{L}(\Theta; p)$ and that for training the parameter $\Theta$. As illustrated by this figure, the dominant training overhead is from the generator training phase, while optimizing the prompt adds negligible training cost. Considering that POQD also yields significant performance gains as Table 2 shows, these findings thus highlight both the effectiveness and efficiency of POQD.

We also report the inference time of POQD in Figure 5. Similar to the breakdown in Figure 4, the total inference time is decomposed into three components, including the generator model inference time, retrieval time, and the time spent on decomposing queries. As illustrated in Figure 5, the model inference time contributes the largest portion of the overall inference overhead, significantly exceeding the query decomposition time. This finding indicates that incorporating query decomposition does not adversely impact the overall inference speed.

### 5.3. Ablation studies

We also perform a series of ablation studies to evaluate the effect of the hyperparameters and the superiority of POQD under various configurations with the WebQA dataset in the text QA task.

**Effect of** $\alpha$ We also vary the value of $\alpha$ in Algorithm 1 to evaluate its effect on the training loss, which produces Figure 6. In this figure, the prompt used for decomposing queries is updated three times by Algorithm 1 (indicated by the inverted triangle symbols). As this Figure suggests, if $\alpha$ is too large (say $\alpha$=0.05), POQD would struggle to find a suitable $p^{\text{new}}$ in Algorithm 1, thus causing the underfitting

| Dataset | Image QA | | | Text QA | | |
|---|---|---|---|---|---|---|
| | WebQA | MultiModalQA | ManyModalQA | WebQA | MultiModalQA | ManyModalQA |
| | Hit@1 | Hit@1 | Hit@1 | Hit@2 | Hit@2 | Hit@2 |
| Dense retrieval | 41.38 | 50.00 | 27.38 | 43.52 | 66.44 | 49.25 |
| ColBERT | 11.98 | 25.22 | 16.30 | 50.72 | 40.92 | 40.37 |
| ColBERT-orig | 38.95 | 36.96 | 21.05 | 52.16 | 79.89 | 87.07 |
| S-QD | 40.93 | 52.61 | 28.15 | 48.56 | 56.17 | 68.07 |
| U-QD | 33.89 | 51.74 | 26.86 | 46.04 | 45.21 | 67.89 |
| ICL-QD | 39.39 | 54.34 | 27.76 | 41.37 | 71.43 | 85.14 |
| ICLF-QD | 41.69 | 51.74 | 27.89 | 51.80 | 69.76 | 66.49 |
| POQD | **42.33** | **58.26** | **28.67** | **53.24** | **80.58** | **92.35** |

*Table 1.* Retrieval Accuracy on QA datasets

| Dataset | Image QA | | | Text QA | | | |
|---|---|---|---|---|---|---|---|
| | WebQA | MultiModalQA | ManyModalQA | WebQA | MultiModalQA | ManyModalQA | StrategyQA |
| w/o RAG | 80.31 | 16.52 | 30.98 | 56.47 | 40.36 | 32.28 | 58.76 |
| Dense retrieval | 81.43 | 46.09 | 34.12 | 59.35 | 59.36 | 41.25 | 61.43 |
| ColBERT | 82.14 | 42.61 | 29.80 | 60.79 | 53.95 | 22.08 | 62.45 |
| ColBERT-orig | 81.96 | 49.13 | 32.29 | 61.14 | 61.73 | 77.66 | 65.50 |
| S-QD | 81.73 | 48.70 | 35.82 | 58.99 | 54.92 | 62.62 | 73.33 |
| U-QD | 82.26 | 47.73 | 33.73 | 60.79 | 49.24 | 60.95 | 71.11 |
| ICL-QD | 82.37 | 46.78 | 34.70 | 58.63 | 61.86 | 76.69 | 60.70 |
| ICLF-QD | 81.54 | 46.55 | 34.77 | 60.42 | 63.52 | 60.07 | 72.34 |
| POQD($\tau$=3) | **82.83** | **61.74** | **37.92** | **62.22** | **68.10** | **81.27** | 75.55 |

*Table 2.* End-to-End QA (Exact Match) Accuracy. We bold the best and underline the second best accuracy number respectively

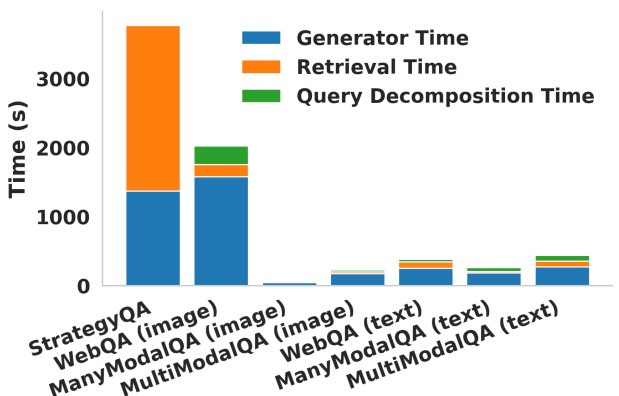

*Figure 5.* Overall inference time of POQD

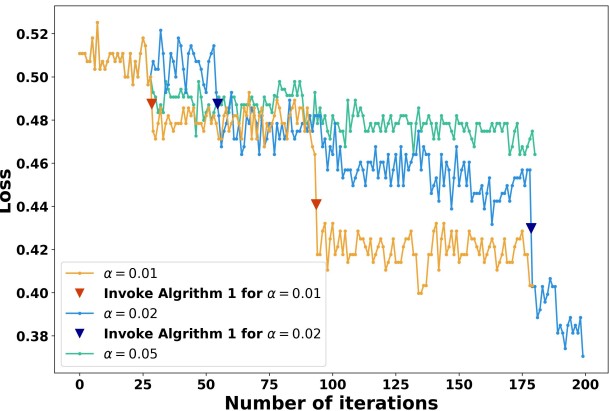

*Figure 6.* Effect of $\alpha$ on the training loss on the WebQA dataset in text QA

issue. In contrast, if $\alpha$ is too small (say $\alpha$=0.01), POQD converges much slower than our default with $\alpha$=0.02. Hence, the default configuration of $\alpha$, i.e., 0.02, can balance the convergence speed and the final performance. In addition, with $\alpha$=0.02, the training loss decreases smoothly throughout the training process, exhibiting no abrupt spikes.

**Effect of $\tau$**  We measure the training loss $\mathcal{L}(\Theta; p)$ and the total training time by varying $\tau$ between 0 to 5, which is plotted in Figure 7. Notably, the performance trend exhibited in this figure matches the analysis in Section 4.4, i.e., larger $\tau$ leading to longer training time but better per-

formance. As this figure suggests, configuring $\tau$ as 3 is a reasonable choice since it balances the training efficiency and performance well.

**Effect of using varied LLMs for decomposing queries**
Unlike other methods, ICL-QD, ICLF-QD, and POQD rely on one LLM for generating decomposed sub-queries. Hence, we also compare their performance with alternative LLMs

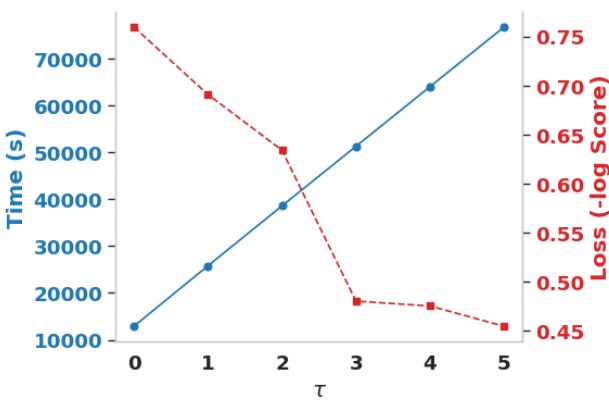

Figure 7. Training time of POQD with varied values of $\tau$

for query decomposition. Specifically, this experiment is conducted by leveraging the GPT-4 model (Achiam et al., 2023) and DeepSeek-V3 (DeepSeek Team, 2024) as the query decomposer, which leads to the results in Table 3. As this table shows, with varied LLMs used for query decomposition, POQD consistently outperforms ICL-QD and ICLF-QD.

| | Top-2 retrieval accuracy | | QA accuracy | |
| | GPT-4 | DeepSeek | GPT-4 | DeepSeek |
| --- | --- | --- | --- | --- |
| ICL-QD | 56.83 | 55.40 | 57.91 | 58.27 |
| ICLF-QD | 49.28 | 48.92 | 56.47 | 57.55 |
| POQD | **58.99** | **55.40** | **59.71** | **60.43** |

Table 3. Performance on the WebQA dataset in text QA with varied LLMs for query decomposition

**Additional ablation studies** Due to space limit, other ablation studies are reported in Appendix D.3, including the study on the effect of the varied number of retrieved items, the filtering step, the varied generator models, and the varied embedding models for retrieval.

### 5.4. Qualitative studies

We expand the example shown in Figure 1 to show the differences between the baseline methods and POQD. Specifically, we report the decomposed sub-queries generated by other baseline methods in Table 4. In comparison to POQD, these baseline methods produce sub-queries that either contain irrelevant information, say the word "what" and "type" generated by S-QD, or miss key information, say "Hong Kong" by U-QD. As a consequence, the most relevant images retrieved with those sub-queries (shown in Appendix D.4) do not match the ground-truth image shown in Figure 1. These retrieval errors can thus be attributed to the unreasonable sub-queries produced by the baseline methods.

| | Decomposed sub-queries |
| --- | --- |
| S-QD | ["What Victoria, Hong Kong", "type buildings"] |
| U-QD | ["Victoria" ,"bulidings"] |
| ICL-QD | ["Victoria Hong Kong", "buildings", "type"] |
| ICLF-QD | ["buildings Victoria, Hong Kong"] |
| POQD | ["Victoria Hong Kong","buildings"] |

Table 4. Performance on the WebQA dataset in text QA by using the Roberta model as the embedding model for retrieval

## 6. Related work

**Multi-Vector Retrieval** Multi-Vector Retrieval (MVR), first introduced by ColBERT (Khattab & Zaharia, 2020), employs a late-interaction mechanism to evaluate query-document similarity. This approach can overcome the representational limitations of dense retrieval methods that use single embeddings for queries and documents. Subsequent works have focused on accelerating retrieval (Santhanam et al., 2022b;a; Gao et al., 2021; Li et al., 2023) or improving score aggregation strategies (Qian et al.). Notably, most solutions decompose queries into individual tokens, leaving the optimization of query decomposition for MVR underexplored.

**LLM-based optimizer** (Yang et al., 2024; Pryzant et al.; Wang et al.) have shown the potential of large language models (LLMs) as a generic optimizer, which aims to search the prompts for a given LLM based on a history of a history of past instructions and performance scores on the training set. Later on, this strategy is further extended for optimizing the configurations of LLM agents (Zhang et al.; Zhao et al., 2024a). This strategy is highly effective since it is free of gradient computation (Lin et al., 2024).

Due to the space limit, we discuss other relevant related works in Appendix C.

## 7. Conclusion

In this paper, we studied how to decompose queries into sub-queries for multi-vector retrieval such that the performance of retrieval-based systems, in particular, RAG systems is optimized. To solve this problem, we propose to prompt an LLM to decompose queries into sub-queries and optimize its prompt with an LLM-based optimizer. We further propose an efficient end-to-end training algorithm. Extensive experiments on a variety of RAG-based QA benchmark datasets demonstrate the effectiveness and efficiency of our method.

## Acknowledgements

This work is supported by "The Fundamental Research Funds for the Central Universities, Peking University". We are grateful to the GPU Cluster support with AIBD platform from Fundamental Industry Training Center of Tsinghua University.

## Impact Statement

This paper presents work whose goal is to enhance the end-to-end performance of the retrieval-based systems, in particular, the Retrieval-Augmented Generation (RAG) systems, through optimizing their retrieval process. This is achieved by utilizing multiple-vector retrieval and optimizing the way of decomposing queries into sub-queries. We believe that the proposed method could be seamlessly employed to enhance the performance of arbitrary RAG systems in a lightweight manner, but without imposing any negative impacts.

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

# A. Theoretical analysis

The proof of Theorem 4.4 depends on the following essential Lemma from prior studies.

**Lemma A.1** (Linear convergence). *For a model parameterized by $\Theta$ satisfying the $\mu$-PL\* condition and the $L$-smoothness, suppose it is trained with the gradient descent method and converges to $\Theta^*$, then the following inequality defined on the loss function $\mathcal{L}(\Theta)$ holds for the model parameter at the $k_{th}$ iteration (denoted by $\Theta_k$):*

$$\mathcal{L}(\Theta_k) - \mathcal{L}(\Theta^*) \le (1 - \frac{\mu}{2L})^k (\mathcal{L}(\Theta_0) - \mathcal{L}(\Theta^*)).$$

*The above inequality indicates that the model $\Theta$ converges at a linear rate.*

*Proof.* First, according to the $\mu$-PL\* condition shown in Assumption 4.1, i.e., $\|\mathcal{L}(\Theta_t)\|_2^2 \ge \mu\mathcal{L}(\Theta_t)$, the following inequality holds:

$$\|\mathcal{L}(\Theta_t)\|_2^2 \ge \mu\mathcal{L}(\Theta_t) \ge \mu(\mathcal{L}(\Theta_t) - \mathcal{L}(\Theta^*)). \tag{4}$$

Then according to the smoothness property of $\mathcal{L}(\Theta_t)$, the following formula holds:

$$\mathcal{L}(\Theta_{t+1}) \le \mathcal{L}(\Theta_t) + \nabla\mathcal{L}(\Theta_t; p)(\Theta_{t+1} - \Theta_t) + \frac{L}{2}\|\Theta_{t+1} - \Theta_t\|_2^2,$$

According to Formula (3), $\Theta_{t+1} - \Theta_t$ in the above formula can be substituted by $-\eta\mathcal{L}(\Theta_t)$, leading to:

$$\mathcal{L}(\Theta_{t+1}) \le \mathcal{L}(\Theta_t) - \eta\|\nabla\mathcal{L}(\Theta_t)\|_2^2 + \frac{L\eta^2}{2}\|\nabla\mathcal{L}(\Theta_t)\|_2^2$$
$$= \mathcal{L}(\Theta_t) - (\eta - \frac{L\eta^2}{2})\|\nabla\mathcal{L}(\Theta_t)\|_2^2.$$

According to Equation (4), the above formula can be transformed to:

$$\mathcal{L}(\Theta_{t+1}) \le \mathcal{L}(\Theta_t) - \eta\|\nabla\mathcal{L}(\Theta_t)\|_2^2 + \frac{L\eta^2}{2}\|\nabla\mathcal{L}(\Theta_t)\|_2^2$$
$$= \mathcal{L}(\Theta_t) - (\eta - \frac{L\eta^2}{2})\|\nabla\mathcal{L}(\Theta_t)\|_2^2$$
$$\le \mathcal{L}(\Theta_t) - (\eta - \frac{L\eta^2}{2}) \cdot \mu(\mathcal{L}(\Theta_t) - \mathcal{L}(\Theta^*))$$

Then by subtracting $\mathcal{L}(\Theta^*)$ on both sides of the above formula, we can obtain:

$$\mathcal{L}(\Theta_{t+1}) - \mathcal{L}(\Theta^*) \le \mathcal{L}(\Theta_t) - \mathcal{L}(\Theta^*) - (\eta - \frac{L\eta^2}{2}) \cdot \mu(\mathcal{L}(\Theta_t) - \mathcal{L}(\Theta^*))$$
$$= [1 - \mu\eta(1 - \frac{L\eta}{2})](\mathcal{L}(\Theta_t) - \mathcal{L}(\Theta^*))$$

Then by setting $\eta = \frac{1}{L}$, $1 - \mu\eta(1 - \frac{L\eta}{2}) = 1 - \frac{\mu}{2L}$, and thus the above formula is bounded by:

$$\le (1 - \frac{\mu}{2L})(\mathcal{L}(\Theta_t) - \mathcal{L}(\Theta^*))$$

$\square$

Recall that according to Assumption 4.1, the training loss satisfies the $\mu$-PL\* condition and $L$-smoothness with arbitrary fixed $p$. Hence, by substituting $\mathcal{L}(\Theta)$ with $\mathcal{L}(\Theta; p)$, the following inequality holds for $\mathcal{L}(\Theta; p)$ when $\Theta$ is updated with gradient descent:

$$\mathcal{L}(\Theta_k; p) - \mathcal{L}(\Theta^*; p) \le (1 - \frac{\mu}{2L})^k (\mathcal{L}(\Theta_0; p) - \mathcal{L}(\Theta^*; p)). \tag{5}$$

Then we formally provide the proof to Theorem 4.4 as follows.

*Proof.* With a fixed prompt $p^{\text{old}}$ (or $p^{\text{new}}$ resp.), suppose the model parameter at the $t_{th}$ iteration of the training process shown in Line 4 of Algorithm 2 is $\Theta_t^{\text{old}}$ (or $\Theta_t^{\text{new}}$ resp.). By representing $A = 1 - \frac{\mu}{2L}$, Equation (3) can be rewritten as:

$$\mathcal{L}(\Theta_t^{\text{old}}; p^{\text{old}}) - \mathcal{L}(\Theta^*(p^{\text{old}}); p^{\text{old}}) = \mathcal{L}(\Theta_t^{\text{old}}; p^{\text{old}}) - G_{\text{old}}$$
$$\leq A^t[\mathcal{L}(\Theta_0^{\text{old}}; p^{\text{old}}) - \mathcal{L}(\Theta^*(p^{\text{old}}); p^{\text{old}})] = A^t[\mathcal{L}(\Theta_0^{\text{old}}; p^{\text{old}}) - G_{\text{old}}],$$

in which we use $G_{\text{old}}$ to represent $\mathcal{L}(\Theta^*(p^{\text{old}}); p^{\text{old}})$ and substitute the ratio $1 - \frac{\mu}{2L}$ with a constant $A(0 < A < 1)$

Then when $t = \tau$, the above formula is transformed into:

$$\mathcal{L}(\Theta_\tau^{\text{old}}; p^{\text{old}}) - G_{\text{old}} \leq A^\tau[\mathcal{L}(\Theta_0^{\text{old}}; p^{\text{old}}) - G_{\text{old}}]. \tag{6}$$

Recall that when $p^{\text{old}}$ is updated to $p^{\text{new}}$, the training loss is reduced by $\alpha$. Therefore, the following formula holds:

$$\mathcal{L}(\Theta_\tau^{\text{old}}; p^{\text{old}}) - \mathcal{L}(\Theta_\tau^{\text{old}}; p^{\text{new}}) \geq \alpha,$$

which can be integrated into Formula (6), resulting in:

$$\mathcal{L}(\Theta_\tau^{\text{old}}; p^{\text{new}}) + \alpha - G_{\text{old}} = \mathcal{L}(\Theta_0^{\text{new}}; p^{\text{new}}) + \alpha - G_{\text{old}} \leq A^\tau(\mathcal{L}(\Theta_0^{\text{old}}; p^{\text{old}}) - G_{\text{old}}). \tag{7}$$

By subtracting $G_{\text{old}}$ and adding $G_{\text{new}}$ on the left side of the above formula, we can obtain:

$$\mathcal{L}(\Theta_0^{\text{new}}; p^{\text{new}}) - G_{\text{new}} + G_{\text{new}} - G_{\text{old}} + \alpha \leq A^\tau(\mathcal{L}(\Theta_0^{\text{old}}; p^{\text{old}}) - G_{\text{old}}).$$

Since $G_{\text{new}}$ minimizes $\mathcal{L}(\Theta; p_{\text{new}})$ for any $\Theta$, this means that $\mathcal{L}(\Theta_0^{\text{new}}; p^{\text{new}}) - G_{\text{new}} \geq 0$. This thus can be leveraged to lower bound the left side of the above formula, leading to:

$$G_{\text{new}} - G_{\text{old}} + \alpha \leq A^\tau(\mathcal{L}(\Theta_0^{\text{old}}; p^{\text{old}}) - G_{\text{old}}).$$

By leveraging the fact that $\mathcal{L}(\Theta_0^{\text{old}}; p^{\text{old}}) \leq M$ and $G_{\text{old}} > 0$, then the above formula is further bounded by:

$$G_{\text{old}} - G_{\text{new}} \geq \alpha - A^\tau M.$$

$\square$

# B. Additional experimental setups

**Datasets and models** As introduced in Section 1, we primarily conduct experiments on RAG-based QA tasks covering both image QA and text QA tasks, in which we evaluate both the retrieval performance and end-to-end QA accuracy. Specifically, we employ **Web Questions** (**WebQA**) (Berant et al., 2013; Chang et al., 2021), **MultiModalQA** (Talmor et al.), **ManyModalQA** (Hannan et al., 2020) and **StrategyQA** (Geva et al., 2021a) dataset for experiments. The WebQA dataset consists of Image QA dataset (Chang et al., 2021) and Text QA dataset (Berant et al., 2013). Among these datasets, WebQA, MultiModalQA and ManyModalQA contain questions that may require retrieving data from different modalities. Hence, for image QA, in particular, for WebQA, MultiModalQA and ManyModalQA dataset, we only select questions requiring retrieving images while for text QA, we only select questions requiring text documents as input. Among these datasets, StrategyQA is employed for multi-hop QA while others are used for single-hop QA.

Regarding the retrieval model for text QA, we employ the pre-trained model developed by (Santhanam et al., 2022b) for ColBERT, which fine-tunes the Bert model (Devlin, 2018) on the MSMARCO dataset (Nguyen et al., 2016). Since ColBERT decomposes queries at the token level, which may not be suitable for decomposed sub-queries, we thus employ the Sentence-Bert model (Reimers, 2019) for encoding sub-queries and corpus for other baseline methods as well as POQD. On the other hand, for image QA, the CLIP model (Radford et al., 2021) is employed to embed text queries and images.

For the generator models, we leverage Llama3.1-8B (Dubey et al., 2024) and Llava-v1.5-7B (Liu et al., 2024) as generators for single-hop text QA and image QA respectively. In the experiments for these tasks, only the generator models are

fine-tuned in our experiments for baseline methods while POQD jointly fine-tunes the generator model and optimizes the prompt for query decomposition. On the other hand, regarding multi-hop text QA, i.e., StrategyQA dataset, we follow the state-of-the-art (Xu et al., 2024) to utilize frozen GPT-4 (Achiam et al., 2023) model and merely replace its default retrieval model, i.e., ColBERT, with other baseline and POQD.

Also, to facilitate multi-vector retrieval, it is essential to construct multiple vectors appropriately for each document or image in the corpus. Since queries are decomposed into sub-queries composed of multiple tokens, it is thus not appropriate to follow (Khattab & Zaharia, 2020) to embed individual tokens in documents, which is even impossible for images in the context of image QA. Hence, for text QA, except ColBERT, we embed individual sentences for other baseline methods and POQD while for image QA, we segment each image into superpixels with existing image segmentation tools, say, SLIC (Achanta et al., 2010), and embed these superpixels with CLIP model.

## C. Additional related work

**Question decomposition in Question Answering** How to decompose queries or questions has been extensively studied in the Question-Answering (QA) literature but from a different perspective. Specifically, various strategies have been devised to decompose complex questions into multiple sub-questions to facilitate multi-hop QA. Indeed, these strategies could be adapted for decomposing queries for multi-vector retrieval. For instance, a series of works (Xue et al., 2024; Yang & Zhu, 2021; Zhou et al., 2022; Zhu et al., 2023; Guo et al., 2022; Wu et al., 2024a; Geva et al., 2021b; Khot et al., 2021) train a sequence-to-sequence model in a supervised manner over a set of training queries with human-annotated sub-queries such that this model can generate decomposed sub-queries directly. Considering the lack of human annotations for sub-queries in most data, one can either train a query decomposition model in an unsupervised way (Perez et al., 2020) or employ heuristics (Jiang et al., 2022; Gandhi et al., 2022; Yang et al., 2023) (say syntax rules (Yang et al., 2023)), or perform in-context learning with LLMs (Li et al., 2024; Pereira et al., 2023; Niu et al., 2023; Ye et al., 2023; Xue et al.; Wu et al., 2024b; Chen et al., 2024; Bhattacharya et al., 2023; Liao et al., 2024; Khot et al.; Press et al., 2023; Zhang et al., 2024) for query decomposition. To further enhance the decomposition performance, one can optionally collect feedback on the decomposed sub-queries and incorporate it into the above in-context learning-based methods. Typical feedback includes confidence scores (Qi et al.), quality scores provided by a powerful enough LLM (Gao et al., 2024), or relevance scores between the generated answers and an expected topic (Sidhoum et al., 2024). Some other works even model the target sub-queries as latent variables (Huang et al., 2021; Zhu et al., 2023). However, to our knowledge, none of these solutions determine sub-queries for optimizing the downstream retrieval-based task performance.

## D. Additional experimental results

### D.1. Prompts used for ICL-QD, ICLF-QD and POQD

We present the prompts used for query decomposition in ICL-QD and ICLF-QD in Figure 8 and Figure 9, respectively. For the latter one, we reuse the prompt from (Qi et al.).

Regarding the prompts used for query decomposition in POQD, we present what prompts are used as the input for the LLM optimizer and that for the LLM-based query decomposer in Figure 10, which also displays how these prompts evolve in the first four iterations of Algorithm 1.

*Figure 8.* The prompt used for query decomposition in ICL-QD

### D.2. Comparison against the query rewrite strategy

There is an emerging trend in the literature for rewriting user queries to maximize the RAG performance. Considering that both POQD and this line of work aim to manipulate the user queries for performance enhancement, we thus also compare POQD against one of such representative work (Ma et al., 2023). This experiment is conducted on the WebQA dataset in

> Imagine that you are a thoughtful and logical problem solver. You will get a question. However, this question is too complex or lacking information to answer. You need to break down the original question into several simpler subquestions to help the information retrieval system retrieve the relevant information. Important: Do not use pronouns or indefinite pronoun phrases in generated questions. The questions asked must be self-contained questions. Each question can contain only one parameter. Don't just ask yes/no questions."
> "For example, Question: Could the Great Wall of China connect the Dodgers to the White Sox? Note: The raised question has to be a self-contained question. Do not use pronouns or indefinite pronoun phrases in the generated questions. Copy context from the original question if needed. Deep Questions: 1. What is the most commonly cited figure for the total length of the Great Wall? 2. What is the straight-line distance between Chicago, Illinois, and Los Angeles, California?

*Figure 9.* The prompt used for query decomposition in ICLF-QD

text QA, which produces results in Table 5. Note that since the query rewrite method still relies on the traditional dense retrieval strategy to retrieve relevant items, it thus suffers from poor retrieval performance, which eventually underperforms POQD by a large margin concerning the end-to-end QA accuracy.

|  | Top-2 retrieval accuracy | End-to-end QA accuracy |
|---|---|---|
| Query rewrite | 28.42 | 52.16 |
| POQD | **53.96** | **61.87** |

*Table 5.* Comparison between POQD and query rewrite strategy

### D.3. Additional ablation studies

We include additional ablation studies in this section. Considering the generally poor performance of ColBERT compared to ColBERT-orig as reported in Section 5, we ignore the results of ColBERT below.

**Effect of the varied number of retrieved items**   We varied the number of retrieved Top-K relevant items in the retrieval process between 1 and 5 on the WebQA dataset in text QA, which leads to the results in Table 6. These results again show that with varied numbers of relevant items, POQD consistently beat other methods regarding the QA accuracy numbers.

|  | 1 | 2 | 5 |
|---|---|---|---|
| w/o RAG | 56.47 | 56.47 | 56.47 |
| Dense retrieval | 58.63 | 59.35 | 61.51 |
| ColBERT-orig | 59.70 | 61.14 | 63.66 |
| S-QD | 58.99 | 58.99 | 62.23 |
| U-QD | 58.27 | 60.79 | 62.23 |
| ICL-QD | 57.91 | 58.63 | 59.35 |
| ICLF-QD | 59.71 | 60.42 | 62.58 |
| POQD | **61.15** | **62.22** | **64.03** |

*Table 6.* End-to-end QA accuracy on the WebQA dataset in text QA by varying the number of retrieved relevant documents in the retrieval process

**Effect of the filtering step**   Considering that LLMs may produce irrelevant tokens during the query decomposition process, POQD filters out these irrelevant tokens in the generated sub-queries. We therefore further evaluate how this filtering step influences the performance of POQD as well as baseline methods.

**Effect of varied generator models**   We repeat the text QA experiments on the WebQA dataset by replacing its default generator model, Llama3.1-8B, with Qwen2.5 (Xu et al., 2025). The resulting end-to-end QA accuracy is reported in Table 8, which suggests that POQD outperforms the baseline methods regardless of the generator model used.

| | Without filtering step | | With filtering step | |
| --- | --- | --- | --- | --- |
| | Top-2 retrieval accuracy | End-to-end QA accuracy | Top-2 retrieval accuracy | End-to-end QA accuracy |
| S-QD | 41.37 | 58.63 | 48.56 | 58.99 |
| U-QD | 39.20 | 57.91 | 46.04 | 60.79 |
| ICL-QD | 35.97 | 57.55 | 41.37 | 58.63 |
| ICLF-QD | 41.01 | 58.27 | 51.80 | 60.42 |
| POQD | **52.51** | **61.87** | **53.24** | **62.22** |

*Table 7.* Performance on the WebQA dataset in text QA with VS without the filtering step

| | Llama3.1-8B | Qwen2.5 |
| --- | --- | --- |
| w/o RAG | 56.47 | 54.32 |
| Dense retrieval | 59.35 | 57.19 |
| ColBERT-orig | 61.14 | 57.55 |
| S-QD | 58.99 | 51.44 |
| U-QD | 60.79 | 50.72 |
| ICL-QD | 58.63 | 57.91 |
| ICLF-QD | 60.42 | 56.47 |
| POQD | **62.22** | **59.35** |

*Table 8.* End-to-End QA Accuracy with varied generator models on the WebQA dataset in the text QA task

**Effect of varied embedding models for retrieval**   We also compare POQD against baseline methods by using the RoBERTa model (Liu et al., 2019) rather than the Sentence-Bert model as the embedding model for retrieval. The results of this experiment are summarized in Table 9, which again indicate the performance advantage of POQD compared to other methods.

| | Top-2 retrieval accuracy | QA accuracy |
| --- | --- | --- |
| w/o RAG | - | 54.32 |
| Dense retrieval | 22.29 | 58.63 |
| ColBERT-orig | 43.53 | 60.43 |
| S-QD | 22.30 | 58.99 |
| U-QD | 20.86 | 57.91 |
| ICL-QD | 39.57 | 59.71 |
| ICLF-QD | 34.89 | 60.07 |
| POQD | **43.88** | **61.51** |

*Table 9.* Performance on the WebQA dataset in text QA by using the RoBERTa model as the embedding model for retrieval

### D.4. Additional details on qualitative studies

As mentioned in Section 5.4, we decompose the query appearing in Figure 1 using POQD and baseline methods and retrieve images with the decomposed sub-queries. The retrieved images by these methods are shown in Figure 11, which suggests that those images do not match the buildings in Victoria, Hong Kong. Hence, this misalignment between these retrieved images and the ground-truth indicates that the decomposed sub-queries by baseline methods (shown in Table 4) are not reasonable.

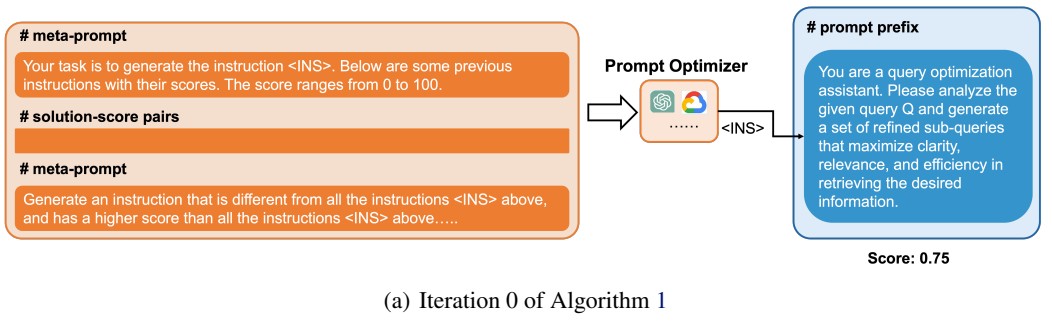

(a) Iteration 0 of Algorithm 1

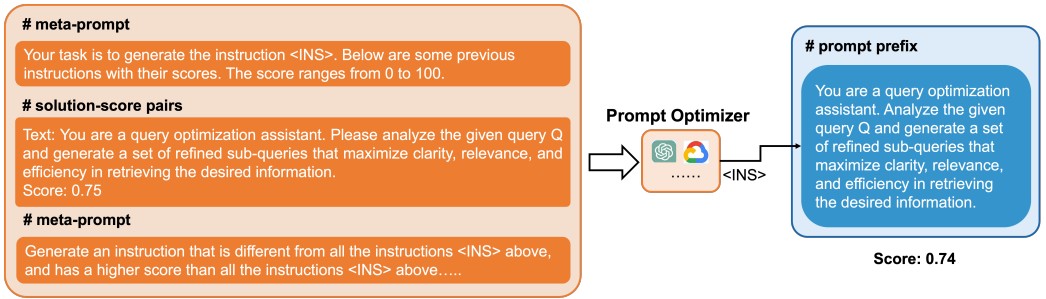

(b) Iteration 1 of Algorithm 1

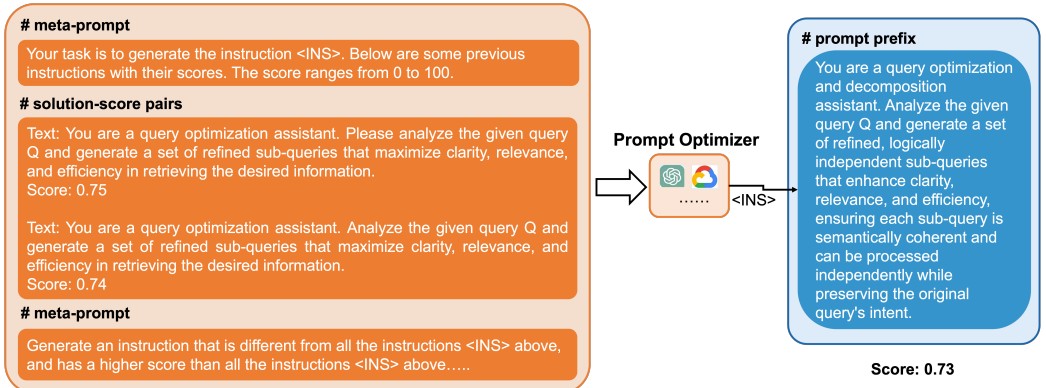

(c) Iteration 2 of Algorithm 1

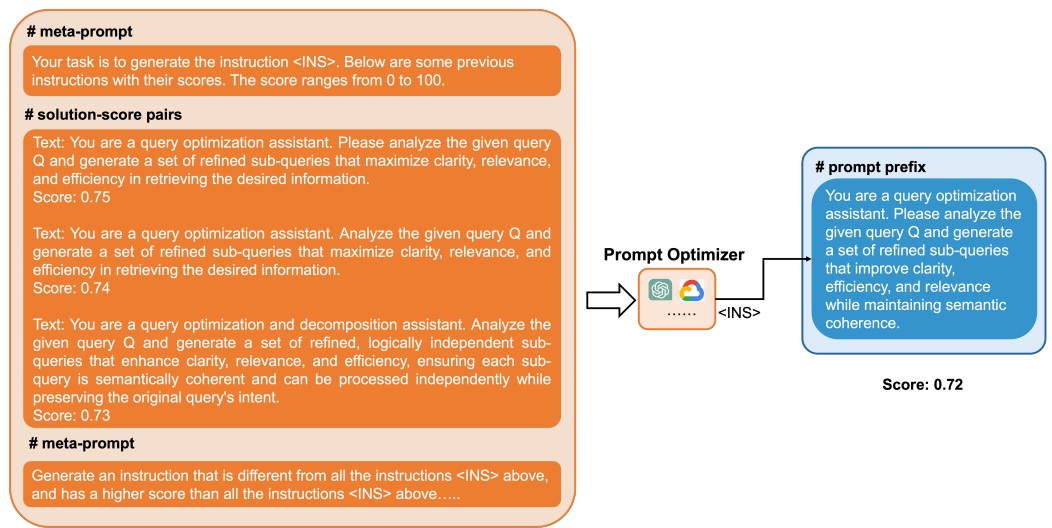

(d) Iteration 3 of Algorithm 1

*Figure 10.* Prompts produced in the first four iterations by Algorithm 1

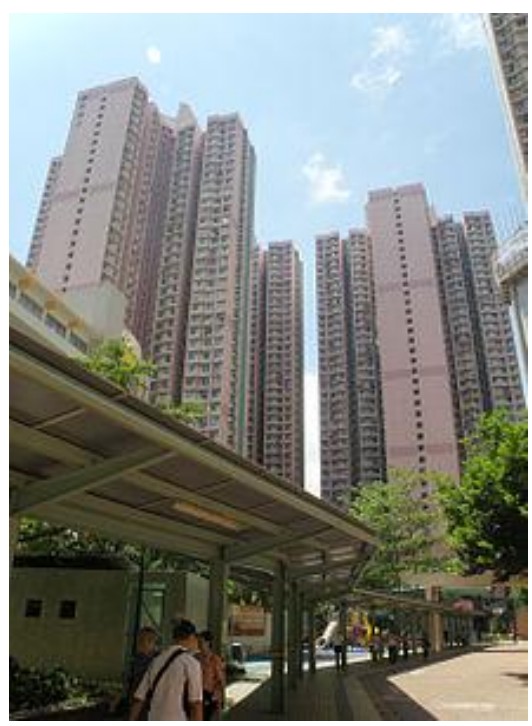

(a) S-QD

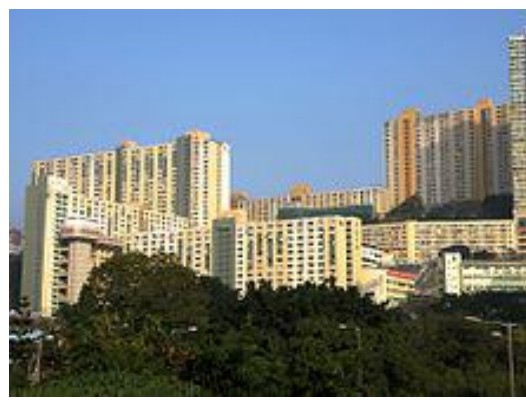

(b) U-QD

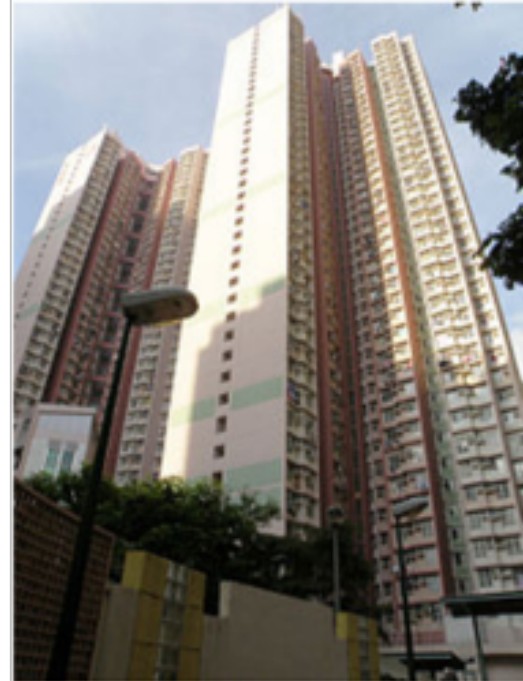

(c) ICL-QD

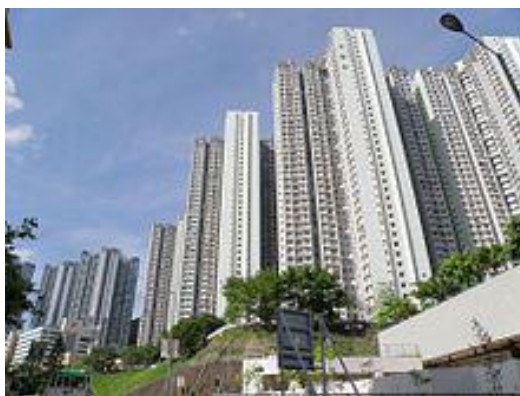

(d) ICLF-QD

*Figure 11.* Retrieved images by using decomposed sub-queries produced by baseline methods

