# OpenReview forum: "POQD: Performance-Oriented Query Decomposer for Multi-vector retrieval"
_ICML.cc/2025/Conference — ICML 2025 poster_

### Official Review · Reviewer_PnGe · 2025-03-13

**Overall Recommendation:** 3

**Summary:**

The paper introduces POQD, a framework for optimizing multi-vector retrieval (MVR) for retrieval-augmented generation (RAG) systems. The key idea is to improve retrieval performance by decomposing a query into sub-queries. POQD uses an LLM in two roles: one acts as a Query Decomposer that splits the input query into candidate sub-queries, and the other functions as a Prompt Optimizer that iteratively refines the prompt guiding the decomposition. It proposes an alternating training algorithm that alternates between optimizing the prompt and training the downstream retrieval model. The paper supports its contributions with experiments across image and text-based QA tasks.

### Update after rebuttal
The rebuttal helped clarify some of my doubts about the experiments, I am raising my score to 3 in light of this. The results in the current manuscript are presented in a confusing manner, and can greatly benefit from a clear presentation.

**Claims And Evidence:**

The central claim of the paper is that better query decomposition leads to significant improvements in multi-vector retrieval performance, it is compared against multiple query decomposition techniques and the simple token-based query decomposition (colbert) but the proposed model uses additional training data which is not clear whether the baselines have access to or not, there are other questions around the experiments which needs further clarification (see Questions below).

**Essential References Not Discussed:**

No.

**Experimental Designs Or Analyses:**

Experiments are performed on QA tasks which is a sound choice for the evaluation of contributed techniques. A better description of the baselines can help in gauging the quality of the baselines used.

**Methods And Evaluation Criteria:**

See"Experimental Designs Or Analyses" comments.

**Other Comments Or Suggestions:**

- Algorithm 1 is hard to understand, many of its details are deferred to later section which makes a linear read of the paper confusing
- Appendix C link in section 6 is not correct
- Figure 5 differs in formatting from the rest of the figures

**Other Strengths And Weaknesses:**

### Strengths
- Paper is well motivated
- Results show significant improvements on ImageQA tasks (although I'm not convinced of the quality of the baselines)

### Weakness
- Limited contributions - the paper's main algorithm is to do an alternate minimization of prompt optimization and retriever training, both of them by themselves are well known techniques, moreover, the theoretical reasoning is very basic assuming very strong assumptions which do not seem fair to make in this setting
- The paper writing can be significantly improved, it's easy to read till motivation section but the presentation can be improved in the method and experiment section

**Questions For Authors:**

1. What are the exact baseline methods in Table 1 & 2 (the baseline description cites multiple works)? Also, are these numbers directly taken from their respective papers? are these baselines also trained on the respective datasets or off the shelf?
2. Why is Colbert much worse than dense retrieval baseline for ImageQA accuracy? Is it because the underlying encoder not trained on image data, in that case underlying encoder should be a capable image retrieval model?
3. Also on TextQA how is it possible that the retrieval accuracy for Colbert is quite high but the final end-to-end accuracy is not?
4. How many iterations does it take on an average to finish the step 1 (prompt optimization for a given retrieval model)?
5. How does query decomposition compare with simple query rewriting?

**Relation To Broader Scientific Literature:**

Query understanding is an important step in performing information retrieval, better query decomposition which is the focus of this paper, is one way to understand a query better which can help the subsequent retrieval pipeline.

**Theoretical Claims:**

Partially.

---

> ### Author Rebuttal · Authors · 2025-03-31
>
> Thank for your comments. Our responses are below:
>
> + About the additional training data:
>
> Indeed, in our experiments, we did not include additional training data to train POQD throughout the paper. During the training process, we train POQD using the same set of training samples as those baseline methods.
>
> + About the baseline method:
>
> As explained in Section 5.1, for S-QD, it aims to train a sequence-to-sequence model to generate decomposed sub-queries. Such training process has been used in many prior works such as  (Zhou et al., 2022; Zhu et al., 2023; Guo et al., 2022; Wu et al., 2024). We thus reuse the code from these papers to train this query decomposition model. For U-QD, we directly use the released unsupervised model for performing query decomposition. For ICL-QD, since it performs query decomposition by prompting LLMs. We thus leverage the prompts from prior works to decompose queries through in-context learning. For ICLF-QD, we follow (Qi et al.) to add retrieval scores as feedback in the prompts for in-context learning.
> The experimental results of these baseline methods are not taken from prior papers. As explained in Appendix C, S-QD, U-QD, ICL-QD and ICLF-QD haven’t been evaluated for decomposing queries for retrieval in the context of RAG. We thus implement these methods in our experiments and obtain these numbers by ourselves.
>
> + About the theoretical analysis:
>
> See our response to reviewer EhPc (About the theoretical analysis)
>
> + About the contribution of the paper:
>
> See our response to reviewer EhPc (About the contribution of the paper)
>
> + About the image retrieval performance of ColBert:
>
> Yes, as mentioned in line 319-321, in the image QA experiments, we leverage the pre-trained CLIP model as the underlying image retrieval model for all methods including ColBert. The reason why Colbert performs worse is that the original version of Colbert has not been applied in image retrieval yet. To mitigate this, we further included the results of Colpali [1] below, which is a variant of ColBert adapted for image retrieval. The results indicate that our method still outperforms Colpali although it is adapted for image retrieval.
>
> |ManyModalQA |Top-1 retrieval acc.| QA accuracy|
> |-|-|-|
> |Colbert| 16.30| 29.80|
> |ColPali| 21.05| 32.29|
> |POQD| **28.67**|**37.92**|
>
> |MultiModalQA |Top-1 retrieval acc.| QA accuracy|
> |-|-|-|
> |Colbert|16.53 |42.61|
> |ColPali|36.96|49.13 |
> |POQD|**48.56**|**61.74**|
>
> + About ColBert text QA results:
>
> See our response to reviewer Nopx (Regarding the retrieval results in Table 1)
>
> + About the number of iterations for step 1:
>
> We configure the number of iterations as 5 by default. If we can find a prompt $p^{new}$ which can lead a training loss smaller than that with $p^{new}$ by up to $\alpha$ within 5 iterations, then we terminate this loop early. Otherwise, we terminate the entire process of updating prompts in Algorithm 2.
>
> + About Algorithm 1:
>
> Some details of this algorithm, particularly the while loop's convergence condition, are deferred to Section 4.3. The loop terminates if the loss with an updated prompt is at least $\alpha$ lower than the initial loss (best\_L in Algorithm 2, equivalent to Algorithm 1's initial loss) within 5 iterations. In other words, convergence occurs when any updated prompt reduces the loss by at least
> $\alpha$ compared to the initial training loss. We will clarify this later.
>
> + About the comparison against the query rewrite strategy:
>
> We further compare POQD against the strategy of performing query rewrite (we follow [2]) on WebQA (text). The following results again show better performance of POQD:
> ||Top 2 retrieval acc.|QA acc.|
> |-|-|-|
> |query rewrite|28.42|52.16|
> |POQD|**53.96**|**61.87**|
>
> [1] Manuel et al. "Colpali: Efficient document retrieval with vision language models."ICLR 2024.
>
> [2] Ma et al, Query Rewriting for Retrieval-Augmented Large Language Models, arxiv

---

### Official Review · Reviewer_sd7s · 2025-03-14

**Overall Recommendation:** 3

**Summary:**

The paper proposes an approach to decompose a query into sub-queries, and the decomposition is optimized to obtain better performance in the end task. The decomposition is performed by using meta-prompting to an LLM. It is expected that the LLM is able to iteratively generate better prompts when the performance measure is provided as feedback. The proposed approach is tested on several datasets, showing generally better performance measures than the baselines.

**Claims And Evidence:**

The main idea of the paper is to iteratively improve the prompt to LLM to generate sub-queries. To this end, an iterative process is designed. Under some conditions, the paper shows that the decomposition improves. This idea sounds very interesting. The experiments provide evidence to show the utility of the process.

**Essential References Not Discussed:**

The main references are cited. They are correctly described.
However, the comparison with them is superficial.

**Experimental Designs Or Analyses:**

The experiments are performed on several datasets. The general design of the experiments are correct.
There is a lack of discussion about the reasons that better performance is obtained with the proposed method, and its differences with the existing baselines, especially on the sub-queries that can be generated.

**Methods And Evaluation Criteria:**

The main idea is attractive. The use of meta-prompt to generate prompts for query decomposition is innovative. The idea sounds intuitive.
However, the paper does not provide sufficient information on what prompts the process may produce. It is unclear that the iterative process will converge. At the end of the iterative process, what type of prompt for query decomposition is generated? Does it correspond to some specific form of prompt? or some pattern for the prompt? If some examples are provided, and better,some analysis about what the iterative process will produce, one can better understand the approach.

The experiments show some advantages of the proposed method over the existing baselines. However, the comparison is mainly quantitative. It is unclear why the proposed method can produce better decompositions. Some analysis about the different behaviors of different approaches would help.

The proposed approach also include a step of filtering of sub-queries with irrelevant tokens. How is this done? How do you determine that a token is irrelevant? How does this filtering important in the whole method? Would part of the gain for the method be generated by this filtering? If this is the case, then the superior performance of your method is not produced solely by the optimized decomposition process. Would a similar filtering process be applicable to other baseline methods? This should be discussed in the paper.

Overall, the evaluation demonstrates the superiority of the proposed method, but there is not sufficient qualitative analysis to allow for better understanding of the reasons.

**Other Comments Or Suggestions:**

General suggestion: perform more details about the decomposition that can be obtained, more analysis about the experimental results and more qualitative comparisons with the baselines.

**Other Strengths And Weaknesses:**

Some statements need further support.
- "Notably, the similarity score between the token “Kong” and the photo of Lee Kuan Yew is exceptionally high,": It is difficult to understand why this is produced. Can you provide a plausible reason for it? This may help understand the nature of the problem.
- The theoretical parts of the paper are more difficult to understand. This is partly due to the fact that some concepts are used without explanation. For example, μ-strongly convex and L-smooth are not explained.
- Sentence BERT is used to encode sub-queries and the corpus: "We thus employ the Sentence-Bert model (Reimers, 2019) for encoding sub-queries and corpus for other baseline methods as well as POQD.". How is this done? Do you consider each sub-query as a sentence and ask sentence-BERT to encode it? How about for the documents? Do you encode each sentence wit it?
- In Fig. 6, why is there a so large retrieval time for WebQA (image)?

**Questions For Authors:**

see the questions above.

**Relation To Broader Scientific Literature:**

The key contribution of the paper is to rely on LLM to decompose queries into sub-queries. The performance-aware optimization is believed to produce a better decomposition than the existing methods. This idea sounds interesting. However, there is a lack of explanation about the process to be fully convinced that the algorithm can indeed produce better decompositions.
The main related methods in the literature are cited.

**Theoretical Claims:**

The paper contains some theoretical results, showing that under some conditions, the iterative optimization process can improve. As recognized by the authors, the conditions will not be satisfied in practice, making the utility of the theoretical results less strong. Despite this, the results are informative.

---

> ### Author Rebuttal · Authors · 2025-04-01
>
> We would thank your comments. Our responses are below:
> + About qualitative examples:
>
> Thanks for pointing out this issue. Indeed, in figure 1, we leverage one example to show the differences between ColBERT, ICL-QD and POQD. We further expand it by reporting the decomposed sub-queries (both before and after filtering step) generated by other baseline methods as follows (the retrieved figures are shown in this [link](https://anonymous.4open.science/r/Example-9717)).
>
> | | before filtering | after filtering |
> |-|-|-|
> |S-QD| ['What is Victoria, Hong Kong known for having in abundance at its waterfront', 'Is  a type of building'] |["What Victoria, Hong Kong", "type buildings"] |
> |U-QD | ['where is Victoria' ,'what is in front of the buliding'] |['Victoria' ,'bulidings'] |
> |ICL-QD| ["historical significance of Victoria Hong Kong", "buildings", "type"] | ["Victoria Hong Kong", "buildings", "type"] |
> |ICLF-QD| [ 'What is the historical significance of colonial buildings in Victoria, Hong Kong ', 'What are the different architectural styles found in Victoria, Hong Kong'] |["buildings Victoria, Hong Kong"] |
>
> Before filtering, the sub-queries from baselines contain many irrelevant tokens, as analyzed below:
> 1. S-QD: Trained on StrategyQA with human-annotated sub-queries. However, when applied to other datasets, distribution shifts introduce irrelevant tokens.
> 2. U-QD: Its candidate sub-queries are collected from datasets (e.g., SQuAD), leading to irrelevant tokens in generated sub-queries.
> 3. ICL-QD/ICLF-QD: Their LLM-generated sub-queries may contain hallucinated irrelevant tokens.
>
> Unlike our method, baselines lack iterative refinement using downstream feedback, causing them to either miss key tokens (e.g., U-QD drops "Hong Kong") or retain unimportant ones (e.g., S-QD keeps "type" and "what"). This leads to incorrect image retrieval and answers. We will add this analysis in the revision.
>
> + About the prompts produced by POQD:
>
> We run Algorithm 1 for four steps and show the prompts and generated prefixes at each step are included in this [link](https://anonymous.4open.science/r/Training-Loss/opt_iter). The LLM optimizer indeed produces variations of query decomposition instructions, searching for variants that minimize the retrieval system’s training loss.
>
> + About the filtering step:
>
> As noted in lines 198-201, we only filter irrelevant tokens to ensure query decomposition correctness. For LLM-based methods (ICL-QD, ICLF-QD), retrieval and QA accuracy remain nearly identical with or without POQD's filtering, as shown in our ManyModalQA (image) ablation studies below.
>
> |Top-1 retrieval accuracy | w/o filtering | w/ filtering|
> |-|-|-|
> |S-QD|  28.15| 27.05|
> |U-QD| 26.86|26.54|
> |ICL-QD| 27.76| 27.51|
> |ICLF-QD| 27.89| 28.37|
>
> |QA score | w/o filtering | w/ filtering|
> |-|-|-|
> |S-QD| 35.82|35.42 |
> |U-QD| 33.73|33.46 |
> |ICL-QD| 34.70|36.37 |
> |ICLF-QD| 34.77| 35.56|
>
> We also removed the filtering step from our method and retested on ManyModalQA (image). The results are shown below, which show that both retrieval and QA accuracy get reduced by 2%-5%, but still higher than baseline methods.
>
> ||Top-1 retrieval accuracy|QA accuracy|
> |-|-|-|
> |POQD (w/o filtering) |28.15 | 36.12|
> |POQD| 28.67 | 37.92 |
>
> + About the theoretical analysis:
>
> See our response to reviewer EhPc (About the theoretical analysis).
>
> + About the example in Section 2.1:
>
> As explained in Appendix B, image retrieval in image QA involves segmenting images into patches and encoding each of them. Due to the maxsim operation, the similarity between a sub-query and an image depends on the most similar patch to this sub-query. In Section 2.1’s example, CLIP considers Patch A [black-filled](https://anonymous.4open.science/r/Training-Loss/kong.png) most similar to "kong," while the ground-truth image’s Patch B [buildings](https://anonymous.4open.science/r/Training-Loss/kong2.png) is less similar. Since "kong" refers to a gorilla-like monster (see [Wiki](https://en.wikipedia.org/wiki/King_Kong)), the Patch A yields unrealistically high similarity, ranking Lee Kuan Yew’s image above the ground truth. We will clarify this in the revision.
>
> + About query and document encodings:
> Yes, we encode sub-queries using pre-trained models like Sentence-BERT. For documents, we encode each sentence similarly for all methods except ColBERT to ensure fair comparison (lines 311-321). ColBERT inherently tokenizes queries and documents, so we use its default setup. But we also test it on WebQA by processing documents like other methods. The top-2 retrieval accuracy (48.92) and QA score (59.71) under this setup are worse than ColBERT's default (61.15 and 60.79 resp.).
>
> + About the retrieval time in Figure 6:
> The retrieval time on WedQA is much smaller than the generator time. You may mean the longer retrieval time on StrategyQA, which as noted in lines 307-311, is due to multi-hop QA requiring repeated reasoning and retrieval per question. We will add this explanation to Section 5.2.

---

> > ### Comment · Reviewer_sd7s · 2025-04-04
> >
> > Thanks for answering the questions.
> > The question about filtering still remains. The filtering removes some irrelevant tokens. How is token's relevance (or irrelevance) determined?
> > About the prompts produced by POQD, unfortunately, I can't open the link to see the examples. So, I still don't understand what type of prompts will be generated.
> > For qualitative analysis of the results, do you have some potential reasons that can explain why your method performs better than the baselines?

---

> > > ### Author Response · Authors · 2025-04-05
> > >
> > > Thanks for your responses! First, sorry for the incorrect link for showing the prompts of POQD. The prompts used during the four optimization steps are included in the following links:  [step 1](https://anonymous.4open.science/r/opt_iter/opt_iter_1.png), [step 2](https://anonymous.4open.science/r/opt_iter/opt_iter_2.png), [step 3](https://anonymous.4open.science/r/opt_iter/opt_iter_3.png) and [step 4](https://anonymous.4open.science/r/opt_iter/opt_iter_4.png).
> > >
> > > Second, regarding the irrelevant tokens, **we regard those tokens appearing in the sub-queries but not from the original queries as irrelevant**. For instance, given the original query is "Victoria Hong Kong has many what type of buildings?" in Figure 1,  before the filtering step, the sub-queries produced by U-QD are 'where is Victoria' and 'what is in front of the buliding'. In those two sub-queries, the tokens "where", "is", "in", "front", "the" are not from the original query, and thus removed in the filtering step, which thus results in 'Victoria' and 'what of bulidings' (sorry for the typos in the rebuttal) as final sub-queries after the filtering step. This aims to remove the potentially negative effects of those irrelevant tokens appearing in the sub-queries. **In addition, as shown in "About the filtering step" above, no matter whether the filtering step is used or not, POQD always outperforms baseline methods**.
> > >
> > > Third, regarding the potential reasons why the baseline methods fail, in addition to the above explanations, we may include the following more intuitive explanations in the revision. **For ICL-QD and U-QD, both of them do not receive any feedback for refining their decomposed sub-queries**. The former relies on the pre-trained LLMs while the latter employs one unsupervised training algorithm to train its query decomposition model. Additionally, **for ICLF-QD and S-QD, although they can receive external feedback for optimizing the query decomposition process, their objectives are not for optimizing the downstream RAG scores**. Specifically, for ICLF-QD, it evaluates the sub-query quality by leveraging another LLM while for S-QD, it evaluates the quality of the sub-queries based on whether it is aligned with the human-annotated sub-queries. However, both the LLM feedback and these human-annotated sub-queries themselves cannot necessarily guarantee better performance of downstream retrieval-based systems. In contrast, our method aims to refine the query decomposition process by optimizing the RAG performance directly, which can thus produce better performance than baseline methods.

---

### Official Review · Reviewer_EhPc · 2025-03-16

**Overall Recommendation:** 3

**Summary:**

The paper presents Performance-Oriented Query Decomposer (POQD), a framework for optimizing query decomposition in retrieval-augmented generation (RAG) tasks, particularly in multi-vector retrieval (MVR). POQD leverages an LLM-driven iterative optimization strategy to generate sub-queries that enhance downstream question-answering (QA) performance. Extensive experiments on RAG-based QA tasks, covering both image QA and text QA have been reported to demonstrate the effectiveness of the proposed framework.

##update after rebuttal
The authors have addressed most of my concerns satisfactorily, I am therefore updating my recommendation to a weak accept.

**Claims And Evidence:**

1. The second contribution of the paper, which pertains to the proposed training algorithm, lacks significant novelty. First, the authors adopt the optimization approach introduced in LLMs as Optimizers for query decomposition, rather than proposing a fundamentally new algorithm. This reliance on existing techniques diminishes the originality of their contribution. Second, the end-to-end training framework is essentially an alternative optimization approach, which is a straightforward and intuitive strategy rather than an innovative methodological advancement. Third, the theoretical analysis presented in the paper is based on critical assumptions that are questionable. Notably, the assumption that the training loss function is strongly convex is highly unrealistic, particularly in the context of complex neural networks, where loss landscapes are typically non-convex and may contain multiple local minima and saddle points. Such an assumption weakens the theoretical foundation of the proposed approach.

2. Furthermore, the third claimed contribution, which concerns the empirical performance of the proposed POQD method, also presents notable issues. While the authors argue that POQD outperforms existing methods, a closer examination of the reported results reveals inconsistencies. Specifically, POQD does not consistently achieve the highest retrieval accuracy across all benchmark comparisons, raising questions about the robustness and generalizability of the approach. The lack of consistent superiority over baselines suggests that the empirical results may not fully substantiate the claimed advantages of POQD.

**Essential References Not Discussed:**

None

**Experimental Designs Or Analyses:**

1. The experimental section of this paper lacks critical implementation details, making it difficult to fully understand and reproduce the proposed approach. Specifically, the implementation of U-QD, ICL-QD, and ICLF-QD is not clearly described. It remains unclear which large language model (LLM) is utilized for both the prompt optimizer and the query decomposer. Additionally, key hyperparameters such as the value of alpha and the number of retrieved items used in the retrieval-augmented generation (RAG) system are not explicitly stated.

2. Moreover, the analysis section lacks an ablation study on the hyperparameter alpha, which is essential for understanding its impact on the overall performance of the proposed method.

3. Furthermore, the experiments do not include an evaluation of the framework’s robustness and generalization ability across different settings. Specifically, there is no empirical analysis demonstrating the performance variations when using different LLMs, retrieval models, or generator models.

**Methods And Evaluation Criteria:**

The experimental evaluation covers both image-based question answering (image QA) and text-based question answering (text QA) scenarios. The evaluation was conducted using four benchmark datasets: WebQA, MultiModalQA, ManyModalQA, and StrategyQA. Among these, WebQA, MultiModalQA, and ManyModalQA include questions that require retrieval from multiple modalities. However, the paper presents image QA results for all three multimodal datasets while reporting text QA results exclusively for WebQA, without providing a clear rationale for this selective reporting. This omission raises concerns regarding the completeness and consistency of the experimental setup, as it remains unclear why text QA was not evaluated across all relevant datasets.

**Other Comments Or Suggestions:**

1. In Assumption 4.3, located on line 274 of page 5, the symbol F(\theta;p) may have been incorrectly represented and should potentially be L(\theta;p).

2. In the Time Analysis section on page 7, the sentence starting with "the generator model is not fine-tuned for this multi-hop QA dataset The results," in line 337 appears to be missing a period.

**Other Strengths And Weaknesses:**

The strengths and weaknesses have been listed above.

**Questions For Authors:**

1. The proposed training algorithm builds upon LLM-based optimization for query decomposition. Could you clarify what specific methodological innovations distinguish your approach from prior work?


2. Could you provide additional insights into the conditions under which POQD performs best and where it struggles? Additionally, have you tested POQD with different LLMs, retrieval models, or generator models to test its robustness? Demonstrating consistent improvements across diverse settings would strengthen the empirical claims.

3. The paper provides image-based QA results on all three multimodal datasets but reports text-based QA results only for WebQA. What was the rationale behind this selective reporting? Would including text QA results for MultiModalQA and ManyModalQA provide additional insights into POQD’s effectiveness across modalities?

4. The impact of the hyperparameter alpha on overall performance is not analyzed in the paper.

5. The paper lacks details regarding the implementation of U-QD, ICL-QD, and ICLF-QD. Could you provide more information on the specific LLMs used for query decomposition and prompt optimizer, as well as key hyperparameters such as the value of alpha and the number of retrieved items in RAG?

**Relation To Broader Scientific Literature:**

The idea of optimizing query decomposition with respect to the final downstream performance might be interesting to broader scientific literature.

**Theoretical Claims:**

I have carefully reviewed the proof. Based on my analysis, given those rigor assumptions, the logical progression and mathematical reasoning appear to be technically sound.

---

> ### Author Rebuttal · Authors · 2025-04-01
>
> Thanks for your comments. Our responses are below:
>
> + About the contribution of the paper:
>
> While we don't propose a novel LLM-based optimization method, our key contribution is recognizing the need to optimize query decomposition for multi-vector retrieval—a critical factor in improving retrieval-based systems. To address this, we adapt an LLM-based optimization strategy for query decomposition and jointly train it with the generator model. We believe this work fits the scope of ICML's application track.
>
> + About the training algorithm:
>
> While our alternative training algorithm may seem straightforward, it could incur higher training overhead compared to optimizing only the RAG generators—a concern for practical applications. However, our theoretical analysis (lines 281-297) shows that with proper hyper-parameters, the added overhead is minimal without compromising optimality, as empirically confirmed in Figure 4. Thus, our algorithm effectively balances efficiency and effectiveness, as highlighted in Contribution 2.
>
> + About the theoretical analysis:
>
> While our theoretical analysis assumes strong convexity, it also holds under the weaker Polyak-Łojasiewicz star (PL*) condition [2]. Prior work [1] suggests that pre-trained, over-parameterized LLMs fine-tuned with GaLore, a variant of LoRA, likely satisfy the PL* condition. Since the model $\Theta$ in the paper is a pre-trained LLM fine-tuned with GaLore, the analysis in [1] applies.
> Assuming the loss function $L(\Theta;p)$ satisfies the $\mu$-PL* condition (instead of $\mu$-strong convexity), a variant of Lemma A.1 yields:
> $L(Θ_k; p) − L(Θ^∗; p) <= (1-\mu/L)^k(L(Θ_0; p) − L(Θ^∗; p))$.
> Similarly, a modified Theorem 4.4 gives:
> $L(Θ^*(p^{old}); p^{old}) − L(Θ^*(p^{new}); p^{new}) >= \alpha - (1-\mu/L)^{\tau} M$.
> These adjustments preserve our conclusion that Algorithm 2 balances efficiency and effectiveness with proper hyper-parameters. We will include this refined analysis.
>
> + About the inconsistent retrieval results:
>
> See our response to reviewer Nopx (Regarding the retrieval results in Table 1)
>
> + About the text QA results:
>
> Indeed, we report both image QA and text QA results for WebQA in Table 1,2. The text QA results on MultiModalQA and ManyModalQA below show that our method consistently outperforms baselines:
>
> |Top-2 retrieval acc.|MultiModalQA|ManyModalQA|
> |-|-|-|
> |Dense retrieval|66.44|49.25|
> |Colbert|79.89| 87.07|
> |S-QD|56.17|68.07|
> |U-QD|45.21|67.89|
> |ICL-QD|71.43|85.14|
> |ICLF-QD|69.76|66.49|
> |POQD|**80.58**|**92.35**|
>
> |QA acc.|MultiModalQA|ManyModalQA|
> |-|-|-|
> |without RAG|40.36|32.28|
> |Dense retrieval|59.36|41.25|
> |Colbert|61.73|77.66|
> |S-QD|54.92|62.62|
> |U-QD|49.24|60.95|
> |ICL-QD|61.86|76.69|
> |ICLF-QD|63.52|60.07|
> |POQD|**68.10**|**81.27**|
>
> + About implementation details:
>
> 1. U-QD: We reuse the code from (Perez et al., 2020).
> 2. ICL-QD & ICLF-QD: Both follow straightforward principles—ICL-QD uses manual prompts for in-context learning, while ICLF-QD adds relevance scores to retrieved documents (see Section 5.1). These prompts will be included in the revision.
> 3. We used the GPT-4 as the prompt optimizer and query decomposer (details in revision).
> 4. Parameters: Default $\alpha=0.02$; retrieved items set to 1 (image QA) and 2 (text QA).
>
> + Ablation studies on WebQA (text):
>
> We first varied the value of alpha and tracked its effect on the training loss is varied across the training process, which is visualized [here](https://anonymous.4open.science/r/Training-Loss/alpha_ablation_study.jpg). If alpha is too large (say alpha=0.05), POQD struggles to find a suitable $p^{new}$ in Algorithm 1, thus causing the underfitting issue. In contrast, if alpha is too small (say alpha=0.01), POQD converges much slower than our default with alpha=0.02. Hence, alpha=0.02 can balance the convergence speed and the final performance.
>
> Ablation on the generator model (from Llama to Qwen2.5):
> |||
> |-|-|
> |Dense retrieval|57.19|
> |Colbert|57.55|
> |S-QD|51.44|
> |U-QD|50.72|
> |ICL-QD|57.91|
> |ICLF-QD|56.47|
> |POQD| **59.35**|
>
> Ablation on the query decomposer (from Llama to GPT4).
> |||
> |-|-|
> |ICL-QD|55.40|
> |ICLF-QD|48.92|
> |POQD|**55.40**|
>
> Ablation on the retrieval model (from sentence-bert to Roberta):
> ||Top-2 retrieval acc.|QA acc.|
> |-|-|-|
> |Dense retrieval|22.29|58.63|
> |Colbert|43.53|60.43|
> |S-QD|22.30|58.99|
> |U-QD|20.86|57.91|
> |ICL-QD|39.57|59.71|
> |ICLF-QD|34.89|60.07|
> |POQD|**43.88**|**61.51**|
>
> Ablation on the number of retrieved items:
> |\# of items|1|5|
> |-|-|-|
> |Dense retrieval|56.31|62.41|
> |Colbert|62.41|69.21|
> |S-QD|51.60|54.23|
> |U-QD|50.49|55.48|
> |ICL-QD|61.58|66.85|
> |ICLF-QD|57.56|60.75|
> |POQD| **62.97**|**69.63**|
>
> These results show the superiority of POQD under various conditions.
>
> [1] Liu, X. H. et al. "On the Optimization Landscape of Low Rank Adaptation Methods for Large Language Models." ICLR 2023.
> [2] Liu, C. et al. "Loss Landscapes and Optimization in Over-Parameterized Non-Linear Systems and Neural Networks." Appl. Comput. Harmon. Anal.

---

### Official Review · Reviewer_Nopx · 2025-03-24

**Overall Recommendation:** 3

**Summary:**

The paper tackles an important problem of jointly optimizing the query decomposition and retreival model for downstream generation task. The query decomposition and embedding model are trained alternatively.

Given an embedding model, the query decomposition is performed using a LLM with the optimization space restricted to prompts. To navigate this space, they use existing ideas from Yang 2024 to have LLM generate better and better prompts.

Given a prompt, the model is trained for some iterations using standard optimization. (e.g. gradient descent)

The authors find that this acheives superior RAG performance on downstream QA tasks -- specifically, the performance is starkly improved on MultiModal data.

**Claims And Evidence:**

A few claims that i was not convinced about are
1.

The prompt decomposition trained the way proposed in the paper achieves superior retreival resutls -- this does not seem to be true generally. Specifically, the Table 1, retreival accuracy (which measures if correct document is retrieved), shows that PQQD does not improve retreival across the board. It has phenomenal improvemenets in MultiModalQA. But it can be quite suboptimal when it comes to TextQA when compared to ColBERT.  It is interesting that despite this, POQD has consistent improvement over all baselines in the downstream task. I believe that the optimization considers if the LLM generating answer with the retrieved documents can generate the correct answer from the correct document or not -- and that seems to tip the scales in favor of POQD. Am i understanding this correctly?

2.

The purpose of theoretical analysis section is unclear to me. On the face of it, it seems like a statement is being made about convergence of the POQD training. However, one most complex pieces of training -- prompt generation is abstracted out in the proof by simply assuming that p_new has lower loss than p_old by amount of $\alpha$ (line 674). Once you assume something like this, along with strong convexity, it is not surprising that the result shows convergence.
However, the loss(p_new) begin less than loss(p_old) is the most tricky part imo. I understand that this is hard to even begin analysing this. But section 4.4 needs to highlight this issue that potentially, POQD can have unstable training where you never get p_new that has lower loss because LLM generating prompt may not satisfactorily navigate the prompt space.

**Essential References Not Discussed:**

I am not well versed with this area of research.

**Experimental Designs Or Analyses:**

I am not well versed with this area of research. The method and evaluation seem reasonable to me based on broader understanding of the field.

**Methods And Evaluation Criteria:**

I am not well versed with this area of research. The method and evaluation seem reasonable to me based on broader understanding of the field.

**Other Comments Or Suggestions:**

1. Line 160. n-sub queries ( do you mean k sub queries)
2. Can you provide more empirical details on training . A few things that might help
     -- how well does prompt scores align with loss.
     -- Can you show us the plots of training loss. I suspect, we can see some unexpected spikes on the prompt update step.

**Other Strengths And Weaknesses:**

[strength]
1. The paper is well written with good examples.

[weakness]
1. The retreival results in table 1 seem contradictory.

**Questions For Authors:**

Please see my concerns above.

**Relation To Broader Scientific Literature:**

I am not well versed with this area of research.

**Theoretical Claims:**

I did not check the correctness. But the result seems okay. Some discussion needs to be added though (see the claims section)

---

> ### Author Rebuttal · Authors · 2025-03-31
>
> We would thank your comments. You can find our responses to your comments below:
>
> + Regarding the retrieval results in Table 1:
>
> We admit that these results look confusing. Indeed, the discrepancy between Table 1 and Table 2 can arise from the fact that we report Top-20 and Top-100 retrieval accuracy while we only retrieve the most relevant image for image QA and the two most relevant documents for text QA. Hence, if we report Top-1 and Top-2 retrieval accuracy instead (as shown below), we can see that our method always performs better than the state-of-the-art:
>
> |Top-1 retrieval accuracy  | ManyModalQA (image) |
> |---|------------------------|
> | Dense retrieval | 27.38 |
> | Colbert |16.30|
> |S-QD|28.15|
> |U-QD|26.86|
> |ICL-QD|27.76|
> |ICLF-QD|27.89|
> |POQD|**28.67**|
>
> |Top-2 retrieval accuracy  | WebQA (text) |
> |---|------------------------|
> | Dense retrieval |52.96|
> | Colbert |52.16|
> |S-QD|48.56|
> |U-QD|46.04|
> |ICL-QD|41.37|
> |ICLF-QD|51.80|
> |POQD|**53.24**|
>
>
> + Regarding the theoretical analysis:
>
> We agree that proving the convergence of POQD is trivial. However, the main message that we want to deliver in Theorem 4.4 (and the following explanations between line 290 and 297) is to analyze under what conditions $p^{new}$ would lead to a better training loss at convergence than that of $p^{old}$. This is crucial in demonstrating that the loss $L(\theta;p)$ is indeed optimized with respect to $p$. Otherwise, it is likely that with the finally derived prompt $p^*$ by Algorithm 2, the converged training loss $L(\theta^*(p^*); p^*)$ is even worse than that with the initial random prompt $L(\theta^*(p^{init}); p^{init})$. This thus invalidates the optimality of $p^*$ derived by Algorithm 2. So to guarantee that this solution is optimal, we thus need to make sure that updating $p^{old}$ to $p^{new}$ can lead to sufficiently large training loss reduction (by $\alpha$) within 5 iterations in Algorithm 1 (which will be clarified in the description of Algorithm 2 in the revision). Otherwise, we terminate updating the prompt.
>
> + Regarding the possibly unstable training issue:
>
> We agree that without identifying a $p^{new}$ that can reduce the training loss by $\alpha$, the training process can be unstable. So as long as we cannot find such a $p^{new}$ within 5 iterations in Algorithm 1, which will be further clarified near line 254 to 257 in the revision, we break the while loop in Algorithm 1 and stop updating $p$ any more. The empirical results in Section 5 indeed demonstrate the performance advantage of our method with this strategy. We would love to elaborate more on this point in the revision.
>
> + Regarding the relationship between the prompt scores and loss:
>
> As we point out in line 180 in the right column, the training loss $L(Θ; p)$ is viewed as the score
>
> + Regarding the plots of training loss:
>
> We included the plot of training loss in this [link](https://anonymous.4open.science/r/Training-Loss/Training_loss.jpg), which shows no spike during the entire training process of Algorithm 2 (including the prompt update process). As mentioned above, we only update $p^{old}$ to $p^{new}$ if the training loss is reduced by at least $\alpha$ within 5 steps (as explained in the response to reviewer PnGe). Otherwise, we terminate the training process. This can thus guarantee the smooth reduction of training loss.

---

> > ### Comment · Reviewer_Nopx · 2025-04-07
> >
> > Thanks for response. I do not have any additional questions. I will maintain my current evaluation for this paper.

---

### Decision · Program_Chairs · 2025-05-01

**Decision:**

Accept (poster)

**Comment:**

This work addresses the problem of query decomposition in multi-vector retrieval (MVR) systems. It argues that the performance of MVR systems is highly dependent on the decomposition of the query into smaller units such as phrases and words. As end-to-end optimization of MVR systems by joint optimization of query decomposer and downstream retrieval-based task is impractical, the work proposes to address the challenge by alternative optimization of the prompt for query decomposition and the downstream models. The work employs the proposed optimization on RAG-based QA tasks and compares the end-to-end task performance with baselines that employ alternative ways of decomposing the query.

While the problem addressed by the work is important, the proposed solution is interesting and the experimental results provide some evidence about the effectiveness of the proposed solution, there are some concerns:

1. The novelty of POQD is limited as it makes a straight forward adaptation of LLM-based
optimizer (Yang et al., 2024) for prompt optimization and a straight forward application of alternative optimization for end-to-end optimization.
2. Some of the experimental results are problematic as noted by reviewer Nopx.
3. Though the theoretical analysis is interesting, it is based on strong assumptions that are not satisfied in practice.
4. Though the experimental results evaluate the effectiveness of the proposed approach, no insights are given on why the proposed solution produces better decompositions as claimed. There is no qualitative analysis that provides deeper insight into specific forms and patterns in the produced prompts.